# KRas-transformed epithelia cells invade and partially dedifferentiate by basal cell extrusion

John Fadul [1], Teresa Zulueta-Coarasa[1], Gloria M. Slattum[2], Nadja M. Redd[3], Mauricio Franco Jin [2], Michael J. Redd[4], Stephan Daetwyler[5], Danielle Hedeen[2], Jan Huisken [6] & Jody Rosenblatt [1✉]

Metastasis is the main cause of carcinoma-related death, yet we know little about how it initiates due to our inability to visualize stochastic invasion events. Classical models suggest that cells accumulate mutations that first drive formation of a primary mass, and then downregulate epithelia-specific genes to cause invasion and metastasis. Here, using transparent zebrafish epidermis to model simple epithelia, we can directly image invasion. We find that KRas-transformation, implicated in early carcinogenesis steps, directly drives cell invasion by hijacking a process epithelia normally use to promote death—cell extrusion. Cells invading by basal cell extrusion simultaneously pinch off their apical epithelial determinants, endowing new plasticity. Following invasion, cells divide, enter the bloodstream, and differentiate into stromal, neuronal-like, and other cell types. Yet, only invading KRas$^{V12}$ cells deficient in p53 survive and form internal masses. Together, we demonstrate that KRas-transformation alone causes cell invasion and partial dedifferentiation, independently of mass formation.

[1] The Randall Centre for Cell & Molecular Biophysics, School of Basic & Medical Biosciences, Faculty of Life Sciences & Medicine, School of Cancer and Pharmaceutical Sciences, King's College London, London, UK. [2] Department of Pediatrics, University of Utah, Salt Lake City, UT, USA. [3] ARUP Laboratories, Salt Lake City, UT, USA. [4] University College London, London, UK. [5] Department of Cell Biology, UT Southwestern Medical Center, Dallas, TX, USA. [6] Morgridge Institute for Research, University of Wisconsin, Madison, WI, USA. ✉email: jody.rosenblatt@kcl.ac.uk

Clinical cancer staging and most models suggest that metastasis arises as a sequential process, where cumulative mutations cause cells to form a primary mass, which then downregulate epithelia-specific genes to enable their detachment and invasion[1]. Yet, metastases can arise independently of primary human tumours[2–6] and mouse models of breast cancer[7,8] and pancreatic cancer[9,10], indicating that metastases occur independently of primary masses. These findings and genetic signatures linked to poor prognosis[11,12] suggest that some oncogenes are intrinsically invasive and that metastasis is not necessarily linked to primary tumours. However, due to our current inability to directly follow stochastic invasion events from epithelia where carcinomas originate, we lack insight into how cells invade. Our research on epithelial cell number homeostasis suggested an unexpected model. Normally, crowding forces trigger live cells to extrude out apically and die by contracting a ring of actomyosin basally, preserving both epithelial function and cell density[13]. However, KRas[V12] and other driver mutations of invasive cancers hijack extrusion, causing cells to instead form masses or aberrantly induce basal cell extrusion (BCE), under the epithelium, potentially enabling their escape[14–16]. Using the transparent zebrafish embryonic epidermis as a model for simple epithelia where carcinomas initiate[17], we directly follow the fate of cells expressing KRas[V12], a driver of poor-prognosis cancers associated with initial stages of tumourigenesis[18–21].

## Results and discussion

We found that mosaically expressing krt4:EGFP-KRas[V12] in the outer epidermal layer caused cells to form masses or extrude, whereas control EGFP-CAAX expression did not (Fig. 1). Masses of 3–40 krt4:EGFP-KRas[V12] cells formed at the zebrafish fin edges, where wild-type cells typically extrude apically[13] (Fig. 1a–c). Strikingly, EGFP-KRas[V12] cells extruded apically and basally at high rates, as scored by hallmark constricting rings, at sites where cells typically divide (Fig. 1d–f, Supplementary Movies 1–3). Extrusion in division zones may be due to high replicative stress and DNA damage[22]. Importantly, BCE occurred at sites distinct from masses in over 600 movies (Fig. 1c, d), with only one cell invading from a mass (Supplementary Movie 4). These surprising findings may have been overlooked previously by studies focusing only on cells invading from masses. EGFP-KRas[V12] but not EGFP-CAAX cells accumulated under the basal epidermal layer, suggesting BCE enables invasion (Supplementary Fig. 1a and Supplementary Movie 5). Internalized EGFP-KRas[V12] cells were independent of misexpression in the notochord, muscle, or melanocytes, typical to transient transgenesis in different zebrafish strains, as UAS:EGFP-KRas[V12] injected into Gal4[−/−] background produced similar rates of misexpressing cells without invaded cells (Supplementary Fig. 1b–d). By contrast, cMyc overexpression, implicated in metastasis[23], did not cause masses, extrusion, or invasion, suggesting that not all oncogenic signalling drives BCE or invasion (Supplementary Fig. 1e, f and Movie 6).

Although EGFP-KRas[V12] BCE caused cells to internalize, many died (Fig. 2a, b), as detected by their fragmentation and rapid engulfment in movies (Supplementary Fig. 2a and Supplementary Movie 7) or by active caspase-3 immunostaining (Supplementary Fig. 2b). While most internalized EGFP-KRas[V12] cells disappear by 5 days post-fertilization (dpf) in a wild-type background, if p53, a pro-apoptotic gene frequently mutated in KRas-driven pancreas, colon, and lung cancers[24–26] is inactivated by either morpholino injection or a loss-of-function M214K mutant[27], internalized cell death significantly decreased, allowing cells to accumulate into large internal masses (Fig. 2b, d, e). p53 expression did not significantly alter KRas[V12]-driven mass formation or BCE rates (Supplementary Fig. 3). By contrast, far

more EGFP-CAAX cells remained within the epidermis in p53 mutants, with none internalized by 5 dpf (Fig. 2c–e), suggesting that p53 mutation alone does not promote BCE and invasion. Thus, while KRas[V12] cells internalize, most die unless they lack functional p53.

Live imaging allowed us to test if BCE can drive cell invasion directly from epithelia. We found that KRas[V12]/p53[MO] cells invade by BCE and then migrate throughout the zebrafish body (Fig. 3a, b, Supplementary Movie 8). These highly motile invaded cells do not represent macrophages that have engulfed EGFP-KRas[V12] cells, as movies labelling DNA indicate that internalized cells do not die (Supplementary Movie 9). When invaded cells do die, fragments that are engulfed lose EGFP signal rapidly (Supplementary Movie 7). Moreover, they do not colocalize with macrophages in movies or immunostained samples (Supplementary Movie 10 and Supplementary Fig. 2a, c). Following invasion, EGFP-KRas[V12]/p53[MO] cells proliferate (Fig. 3c, Supplementary Movie 11, showing two divisions within 15 h). In addition, most EGFP-KRas[V12] (but not control EGFP-CAAX) embryos contained circulating EGFP cells, which frequently impeded blood flow (Fig. 3d, f, g and Supplementary Movies 12–14). Interestingly, circulating EGFP-KRas[V12] cells were also present in p53 wild-type 2 dpf embryos (Fig. 3g), but disappear by 5 dpf (Fig. 2d, e). This may reflect many recent clinical findings that circulating tumour cells arise early in tumorigenesis[10,28].

Remarkably, EGFP-KRas[V12]/p53[MO] cells adopt new morphologies and markers, suggesting new plasticity. Around 50% of embryos contained neuron-like EGFP-KRas[V12]/p53[MO] cells that elongate bi-directionally (Fig. 3e, h, Supplementary Fig. 4a and Supplementary Movie 15), rarely seen in EGFP-CAAX/p53[MO] embryos. While some neuron-like cells express the neuronal marker acetylated tubulin, others localized along endogenous neurons (Fig. 3e, Supplementary Fig. 4b). In addition, ~15% invaded cells express the mesenchymal marker, N-cadherin (Fig. 3i, j). Notably, pancreatic cancer, also driven by KRas/p53 mutations, is predominantly stromal with significant neuronal involvement[29]. Moreover, lineage-tracing studies of mouse pancreatic cancer suggest that stromal cells are partially derived from epithelia[9]. Our data suggest that BCE drives not only cell invasion but can also enable new plasticity.

To determine how cells invading by BCE adopt new fates, we first assessed the epithelial markers, E-cadherin and ZO-1. We found all invaded EGFP-KRas[V12]/p53[MO] cells lacked E-cadherin (340 cells in 25 fish), whereas those remaining in the epidermis retained E-cadherin (900 cells in 25 fish) (Fig. 4a). Interestingly, while most EGFP-CAAX cells remained at the epidermis (Fig. 4b, 13,006 cells in 30 fish), the few that invaded (Fig. 4c, 43 cells in 30 fish) were all E-cadherin-negative, suggesting E-cadherin loss was tightly linked to BCE, rather than to oncogenic mutation per se. ZO-1 was similarly decreased in invaded EGFP-KRas[V12]/p53[MO] cells (Supplementary Fig. 5a, b). E-cadherin expression remained absent in large internal masses in 5 dpf KRas[V12]/p53[mut] larvae (Fig. 4d). While much of EGFP signal within invading KRas[V12] cells is initially pinched off, its re-expression from the keratin 4 promoter suggests that cells invading by BCE continue to express this cytoplasmic epithelial marker as well as KRas[V12] (Fig. 2e). Thus, cells invading by BCE lose cell surface epithelial determinants but retain cytoplasmic ones, with only some adopting mesenchymal features. These findings could address why metastatic tumours can express both epithelial and mesenchymal markers[30,31].

How do invading EGFP-KRas[V12]/p53[MO] cells lose apical epithelial markers? Dynasore treatment suggested that invasion and E-cadherin loss do not require dynamin-dependent endocytosis (Fig. 4e and Supplementary 5c, d). In addition, EGFP-KRas[V12]/p53[MO] embryos rarely expressed the mesenchymal transcription

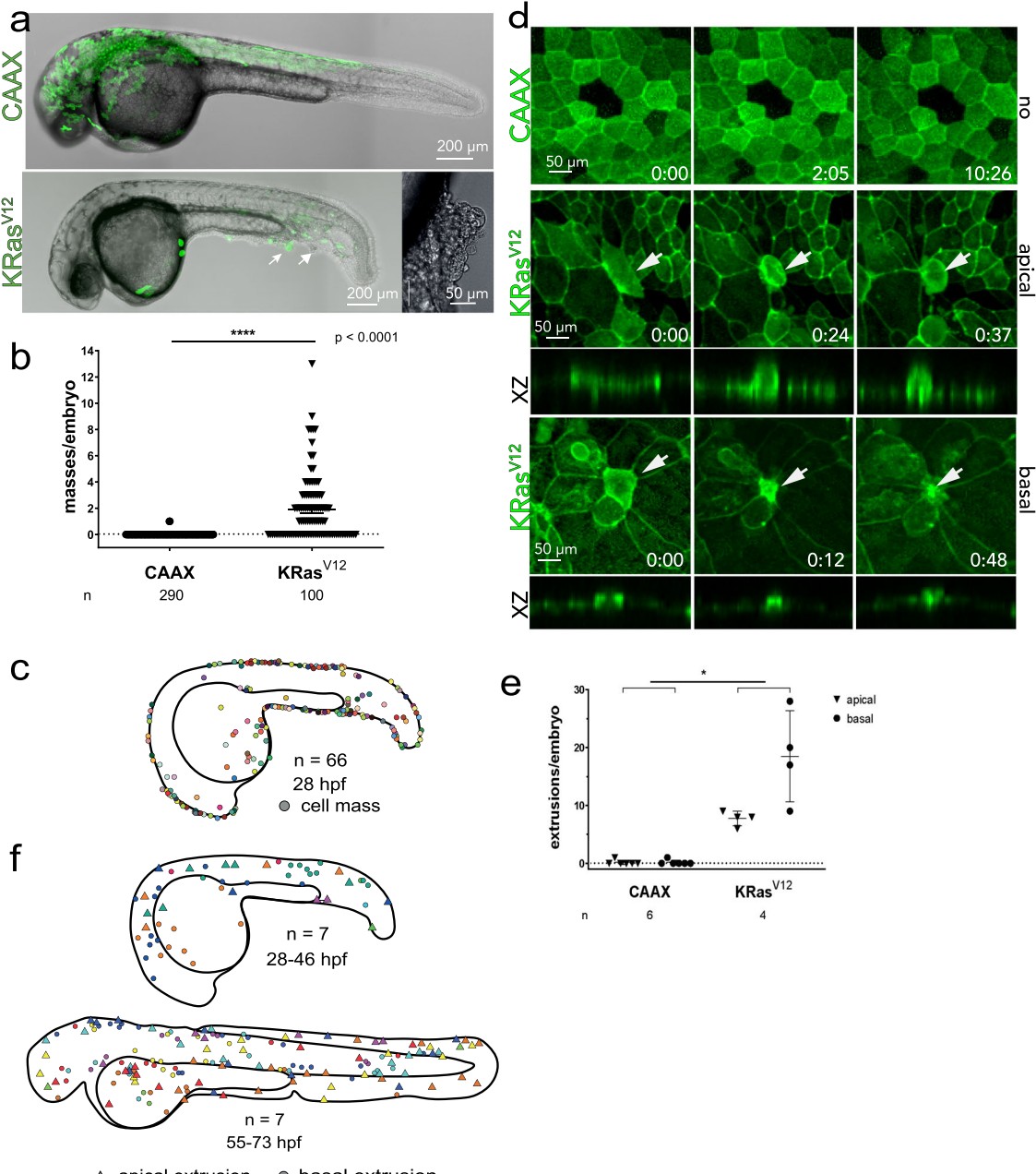

**Fig. 1 KRas^V12 induces formation of epidermal cell masses and basal extrusion at distinct sites. a** 26 hpf wild-type zebrafish embryos expressing *krt4*:EGFP-CAAX or dt-KRas^V12, masses indicated by black arrowheads and inset. **b** Mean of cell masses per embryo ± SEM, $P < 0.0001$ by a Mann–Whitney test, $n =$ embryos. **c** Map of where cell masses occur, as observed at 28 hpf. **d** Stills from time-lapse imaging (hh:mm) of periderm:Lifeact-EGFP mosaically expressing EGFP-CAAX or EGFP-dt-KRas^V12, showing hallmarks of apical and basal extrusion, with XZ sections beneath showing constriction at the base or apex, respectively (white arrows delimit ring constricting). **e** Number of apical and basal extrusions in EGFP-CAAX or dt-KRas^V12 embryos. Data are mean ± SEM, **$P < 0.02$ by a Mann–Whitney test for apical and basal extrusions pooled together. **f** Maps indicating where apical and basal extrusions occur in 28-46 hpf embryos and 55–73 hpf embryos, where different colours represent different fish analyzed, n. Source data are provided as a Source Data file.

factor snai1b in the epidermis or masses, compared to internalized cells, suggesting that BCE does not require snai1b-dependent transcription (Fig. 4f and Supplementary Fig. 5e). While we cannot rule out by in situ hybridization that transient snai1b expression causes transcriptional downregulation of E-cadherin immediately before BCE, we think it unlikely for the following reasons: (i) reported E-cadherin protein perdurance of 3–5 h[32] would predict some internalized cells would retain E-cadherin, (ii) all basally extruding CAAX cells also lose E-cadherin (Fig. 4c), (iii) only ~15% of invading cells express

mesenchymal markers (Fig. 3j), suggesting invasion is independent of mesenchymal gene upregulation, and (iv) some wild-type epidermal cells also express snai1b (Supplementary Fig. 4 c, d), independent of KRas^V12 expression.

To investigate how E-cadherin could suddenly disappear during BCE, we filmed cells expressing EGFP-directly tethered to KRas^V12 (EGFP-dt-KRas^V12) that localizes to the apical membrane[18]. High-resolution imaging revealed that as a cell invades by BCE, the actin ring pinches off its entire apical membrane, releasing the cell underneath the epidermis (Fig. 4g,

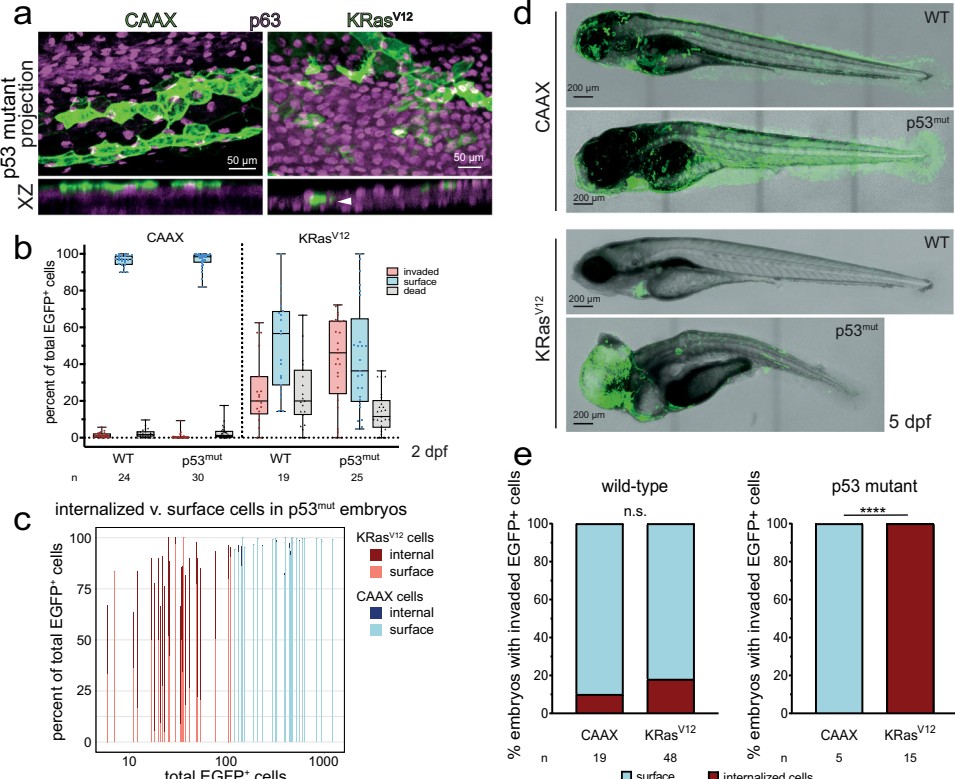

**Fig. 2 p53 loss increases survival of internalized KRas[V12] cells in the zebrafish body. a** Maximum intensity and XZ projections of EGFP-CAAX- or dt-KRas[V12]-injected p53mut 48 hpf embryos. The white arrowhead indicates a KRas[V12] cell internalized beneath p63+ basal keratinocytes. **b** Quantification of invaded vs surface (live + dead) EGFP-CAAX or dt-KRas[V12] cells in wild-type or p53mut embryos, expressed as a percentage of total EGFP+ cells for each embryo, n = embryos, ***P < 0.001 by a Chi-Square test. Data are represented as boxplots where the middle line is the median, the lower and upper hinges correspond to the first and third quartiles, and the whiskers extend from the minimum to the maximum. All data points are shown. **c** Percentage of internalized or surface cells in CAAX or dt-KRas[V12] 48 hpf embryos compared to total EGFP+ cells in each embryo (X-axis, log scale), where each line represents an individual p53mut embryo from (**b**). **d** Wild-type or p53mut larvae at 5 dpf expressing EGFP-CAAX or dt-KRas[V12]. Note that the *cmlc2*:GFP heart cells represent a Tol2 transgenesis marker, unrelated to CAAX or KRas[V12] expression. **e** Percentages of 5 dpf larvae containing surface or invaded EGFP+ cells, ****P < 0.001 by a Fisher exact test. Source data are provided as a Source Data file.

Supplementary Movie16). This severing of the apical membrane explained why all cells invading by BCE experienced concomitant membrane blebbing and loss of green fluorescence before migrating from sites of extrusion (e.g. Supplementary Movies 8, 17, 18 and Fig. 3a). Moreover, E-cadherin colocalizes predominantly with apically localized actin in the periderm here and in other studies[33] and becomes constricted into an apical point above a cell during BCE (Supplementary Fig. 5f and Movie 19 and Fig. 4h), suggesting it is also stripped from the invading cell. While BCE is reminiscent of cytokinesis, Fig. 4h and other images show the nucleus remains beneath the contractile ring, disfavouring asymmetric cell division as an invasion mechanism, seen in Src-expressing cells[34].

To test if invasion and loss of epithelial determinants require BCE, we blocked this process using previously published methods. Because KRas[V12] promotes basal extrusion through increased autophagic degradation of Sphingosine 1-Phosphate, a key signal for apical extrusion, we reported that blocking autophagic flux by ATG knockdown and several inhibitors rescues both S1P and apical extrusion[16]. To reverse autophagy in zebrafish without the lethality intrinsic to genetic methods[35], we used chloroquine to impede KRas[V12]-dependent autophagy and BCE[16]. We find that chloroquine significantly decreased invasion of EGFP-KRas[V12]/p53[MO] zebrafish epidermal cells, compared to vehicle control (Fig. 4i and Supplementary Fig. 5g). In addition, Rockout, a Rho kinase inhibitor that blocks actomyosin

contraction in zebrafish[36,37], which is required for BCE, similarly impaired EGFP-KRas[V12]/p53[MO] cell invasion and E-cadherin loss compared to DMSO control treatment, supporting a role for BCE in both invasion and E-cadherin loss (Fig. 4j and Supplementary Fig. 5h). Together, the tight link between BCE and E-cadherin loss suggests that BCE mechanically drives invasion of KRas[V12] cells and simultaneous loss of apical epithelial determinants. While in our studies, E-cadherin and ZO-1 predominantly localize with cortical, apical actin and are stripped off by BCE, it is important to note that cells invading from epithelia where E-cadherin also localizes basally, such as stratified epithelia[38,39], may not lose their epithelial determinants. In addition, cells could escape by extruding apically into the duct, in which case, we would expect them not to lose apically localized epithelial determinants as they invade. Future work will need to determine if cells invading with E-cadherin lose expression over time through different mechanisms or if the mechanism of invasion impacts later fate and tumour aggressiveness.

Here, we describe a new mechanism driving invasion of KRas[V12]-transformed cells that simultaneously mechanically pinches off the apex, containing most epithelial determinants, and endows escaping cells with new plasticity (Supplementary Fig. 6). In contrast to previous models where cells sequentially downregulate epithelial genes once they accumulate into a mass, we find that KRas-transformation alone, typically associated with early tumourigenesis steps, directly causes invasion through BCE,

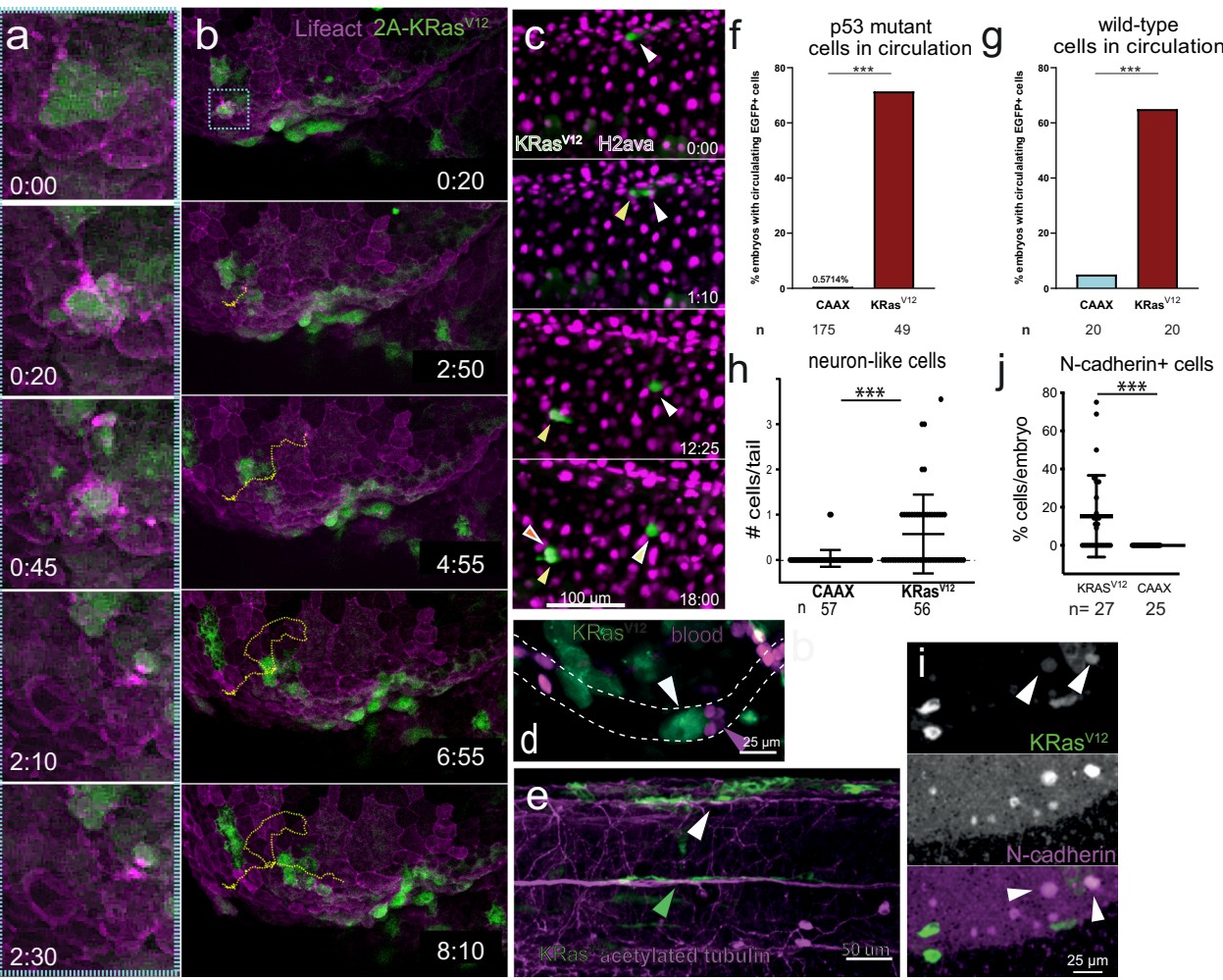

**Fig. 3 KRas$^{V12}$ cells invade by BCE, migrate, divide, and adopt new plasticity. b** Stills from Supplementary Movie 8 of a basally extruding EGFP-T2A-KRas$^{V12}$ cell from a periderm:Lifeact-mCherry reporter line, where (**a**) is a zoom inset (blue) for only the first 2.5 hrs, showing BCE and migration, tracked with dashed yellow line. **c** Stills from Supplementary Movie 11 of an invaded EGFP-KRas$^{V12}$ cell dividing in a *h2afva*:h2afva-mCherry reporter line, where white, yellow, and red arrowheads indicate daughter cells from the two cell divisions. **d** Stills from Supplementary Movie 14 showing an EGFP-KRas$^{V12}$ cell blocking flow of *gata1*:mCherry-labelled blood cells. In all cases, movies were from 24-48 hpf with (hh:mm). Percentage of CAAX or T2A-KRas$^{V12}$ embryos with circulating EGFP+ cells in p53mut (**f**) and wild-type (**g**) zebrafish as mean + SEM, ***$P < 0.001$ by a Fisher Exact test, $n$ = embryos. **e** T2A-KRas$^{V12}$ cells with neuron-like morphology, with white arrowhead indicating co-staining with acetylated tubulin and green arrowhead aligning along a neuron. **h** Neuron-like EGFP-CAAX or T2A-KRas$^{V12}$ cells per n embryos (beneath), ***$P < 0.001$ by a Mann–Whitney test. **i** N-cadherin immunostaining indicating some dt-KRas$^{V12}$ (arrowheads) adopt this mesenchymal marker. **j** Percentage of invaded cells that are N-cadherin+ per embryo, as mean ± SEM, where ***$P < 0.001$ by a Mann–Whitney test. All graphs were from embryos fixed at 48 hpf. Source data are provided as a Source Data file.

independent of primary masses. Whereas p53 and cMyc alterations have been associated with later stages of cancer, they do not drive invasion in our system. We find that KRas-transformation directly causes cell invasion, while p53 mutation enables their survival. While our zebrafish model may endow invading cells with more plasticity than might exist in adult organs, we believe that this ability to view invasion live from native epithelial sites reveals new insight that could account for the intrinsic metastatic and stromal nature of KRas-driven tumours. In addition, understanding how cells invade could shift treatment modalities. For instance, screening for cytoplasmic epithelial markers in circulating tumour cells combined with preventative therapies could ward off metastatic disease.

## Methods

**Molecular biology**. All cloning procedures followed protocols described previously[34] the Invitrogen Gateway Technology Manual, and the Tol2kit wiki (http://tol2kit.genetics.utah.edu/index.php/Main_Page). We cloned the *krt4* promoter sequence into p5E (gift from David Grunwald, University of Utah, Salt Lake City, UT, USA) using the BP Clonase II Enzyme mix (Thermo Fisher Scientific). Using BP recombination, we also cloned into pME the following contructs: human EGFP-dt-KRas$^{V12}$ (isoform 4b) in a pEGFP-C3 backbone (gift from Channing J. Der, University of North Carolina, Chapel Hill, NC, USA), EGFP-CAAX (Tol2kit, gift from Kristen Kwan, University of Utah, Salt Lake City, UT, USA), and mCherry-T2A-cMyc in pUltraHot (gift from Conan Kinsey, University of Utah, Salt Lake City, UT, USA). To generate constructs for microinjection, we recombined p5E-*krt4*, pME-EGFP-dt-KRas$^{V12}$ and p3E-polyA, or p5E-*krt4*, pME-EGFP-CAAX and p3E-polyA, or p5E-*krt4*, pME-mCherry-T2A-cMyc and p3E-polyA, into pDestTol2CG2 using the LR Clonase II Plus Enzyme mix (Thermo Fisher Scientific) to produce *krt4*:EGFP-dt-KRas$^{V12}$, *krt4*:EGFP-CAAX or *krt4*:mCherry-T2A-cMyc, respectively. The EGFP-T2A-KRas$^{V12}$ construct was synthesized and cloned into pUC57 commercially (GenScript), then recombined into pME. Finally, pME-EGFP-T2A-KRas$^{V12}$ was recombined with p5E-*krt4* and p3E-polyA into pDestTol2CG2 to generate the *krt4*:EGFP-T2A-KRas$^{V12}$ for microinjection.

**Zebrafish Use**. All zebrafish embryos and procedures were treated ethically, under approval of licenses issued from both the US Animal Welfare Act., and the UK the Animals in Scientific Procedures Act 1986 and animal use guidelines of the UK Home Office. In addition, all treatments and raising of fish were in compliance with the University of Utah - Centralized Zebrafish Animal Resource and the

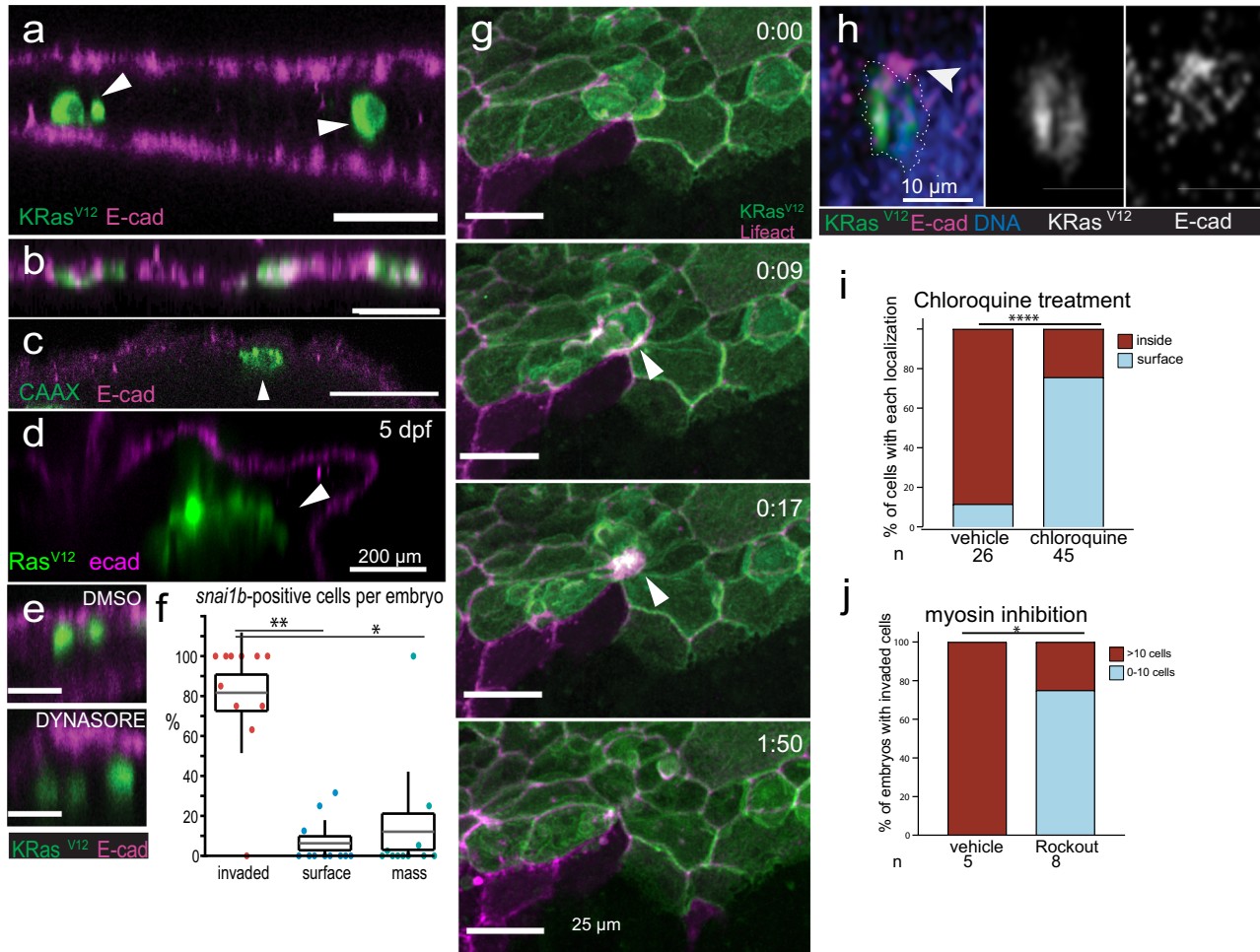

**Fig. 4 KRas$^{V12}$ cells invade by BCE and shed apical epithelial determinants.** XZ sections of 48 hpf EGFP-dt-KRas$^{V12}$ embryos with E-cadherin$^-$ internalized cells (**a**, arrowheads) and EGFP-CAAX embryos, where cells remaining at epidermis and are E-cadherin$^+$ but the few that internalize are E-cadherin$^-$ (**b**, **c**, arrowhead). Those in an internal cell mass (T2A-KRas$^{V12}$, arrowhead) at 5 dpf also lacks E-cadherin (**d**). **e** XZ sections of DMSO or 100 μM Dynasore-treated T2A-KRas$^{V12}$/p53 MO-injected embryos showing internalized cells lack E-cadherin (arrowheads). **f** Percentage of invaded, surface, or mass-associated snai1b+ cells in 14 embryos, *P < 0.05, **P < 0.01 by a Friedman test and Wilcoxon signed-rank test with the Holm-Sidak adjustment for pairwise comparisons, n = 11 embryos. Data are represented as boxplots where the middle (grey) line is the mean, the hinges indicate ±SEM, and the whiskers indicate ±SD. All data points are shown. **g** Stills from Supplementary Movie 16 showing BCE of a EGFP-dt-KRas$^{V12}$ cell in a periderm:Lifeact-mCherry line (arrowheads pointing to ring contracting), clipping off the apex (green). **h** XZ-section of a basally extruding EGFP-T2A-KRas$^{V12}$ cell (dashed) with E-cadherin constricted apically (arrowhead). Analysis of invasion in vehicle versus chloroquine-treated (**i**) or Rockout-treated (**j**) embryos at 48 hpf, ***P < 0.001, *P < 0.05 by a Fisher Exact Test. Source data are provided as a Source Data file.

---

Institutional Animal Care and Use Committee guidelines in compliance with King's College London - Biological Services Unit.

**In vitro transcription.** Transposase and α-bungarotoxin mRNA were in vitro transcribed using the SP6 and T7 mMESSAGE mMACHINE Transcription Kits (Thermo Fisher Scientific), respectively, and purified using NucAway spin columns (Thermo Fisher Scientific). The concentrations of all nucleic acids used were measured using a NanoDrop 1000 (Thermo Fisher Scientific), an EPOCH 2 microplate (BioTek), or a NanoPhotometer N60 (Geneflow) spectrophotometer.

**Microinjections and fluorescence sorting.** 2 nL of a 10-μL injection mix, comprised of 100 ng *krt4*:EGFP-CAAX, 150 ng *krt4*:EGFP-dt-KRas$^{V12}$, *krt4*:EGFP-T2A-KRas$^{V12}$, or UAS:EGFP-KRas$^{V12}$ [17], or 200 ng *krt4*:mCherry-T2A-cMyc + 200 ng transposase mRNA + 1 μL phenol red (Sigma) in nuclease-free dH$_2$O (Ambion), was microinjected into one-cell embryos. Some experiments included 0.2 pmol p53 morpholino (Gene Tools, 5′-GCGCCATTGCTTTGCAAGAATTG-3′) or 25 ng α-bungarotoxin mRNA[35]. Embryos were sorted for expression of transgenes at 1 dpf using a fluorescence dissection microscope, dechorionated with forceps at 1 or 2 dpf or with 1 mg/mL pronase, incubated in E3 with 0.003% N-phenylthiourea (PTU-E3, Merck), and prepared for live imaging or fixed and immunostained. Our studies were not blinded as injected embryos are very easy to

visually distinguish, owing to the development of epidermal cell masses in EGFP-KRas$^{V12}$ embryos (both "dt" and "T2A" versions), absent in the majority of EGFP-CAAX embryos (see Results, Fig. 1A, and Supplementary Fig. 3A). All zebrafish embryos and adults were treated ethically in compliance with our UK Project Licence P946C972B.

**Time-lapse confocal imaging.** The protocol for mounting embryos for live imaging is described[36]. Briefly, dechorionated embryos were anesthetized in 0.02% tricaine in PTU-E3 for 5 min or until no visible movement is observed, then mounted in 0.4–0.8% low-melt agarose as close as possible to the #1.5 glass coverslip within a slide chamber, covered with 0.02% tricaine in PTU-E3, and incubated in a controlled environment chamber at 28 °C and 85% humidity. Timelapse imaging of embryos were done for 18-20 h with 3–7-min time intervals overnight on an Andor Revolution spinning disk confocal microscope (Nikon CFI Plan Apo 20X/0.75 DIC M ∞ /0.17 WD 1.0 objective), a Nikon A1R resonant scanning system (Nikon CFI Plan Apo 20X/0.75 objective), a Leica SP8 white light laser point-scanning confocal microscope, or a Yokogawa CSU-W1/Nikon Eclipse Ti2 spinning disk system equipped with an Andor iXon EMCCD camera. Nikon NIS Elements Advanced Research (v 4.60 & v5.30), Leica Application Suite X (v3.4.2.18368), or Fiji (ImageJ 1.53c) were used for analysis. A Fiji plugin was used to track cells in movies with an arrow[40].

**Time-lapse light sheet imaging.** Selective Plane Illumination Microscopy (SPIM) was used for long-term in toto imaging using one of two custom-built setups: a *multidirectional SPIM (mSPIM)*[37,38] and a *four-lens SPIM* setup[39]. The *mSPIM* was equipped with an UMPlanFL N Olympus 10 × /0.3 NA detection objective, a 488 nm and a 561 nm Coherent Sapphire laser, two Zeiss 10 × /0.2 illumination objectives and two Andor iXon 885 EMCCD cameras. The whole zebrafish embryo was imaged from several angles with 7 × overall magnification and a z-stack spacing of 3 μm every 5–7 min for up to 3 days. To image the whole embryo, several regions were acquired and stitched together using custom image processing plugins[38] in Fiji[40] and maximum intensity projections were generated and registered using SimpleElastix[41] and GUI-based manual rigid registration.

**The four-lens SPIM.** consisted of four identical water-dipping Olympus UMPLFLN 10×/0.3 objectives, two for illumination and two for detection. Two Toptica iBeam smart lasers were externally triggered for alternating double-sided illumination. The laser beam was split 50/50 and directed onto a continuous running galvanometric mirror (1 kHz, EOPC), which pivoted the light sheet and reduced shadowing effects in the excitation paths due to absorption of the specimen[37,38]. Light sheets were generated with cylindrical lenses and projected with telescopes and the illumination objectives onto the focal plane of both detection lenses. The focal planes of the two detection objectives were imaged onto two Andor Zyla sCMOS cameras. The whole embryo was imaged from several angles with a z-stack spacing of 2 μm every 30 s up to 5 min for up to 36 h. A custom LabVIEW (National Instruments) program was implemented to adjust stage positions, stack coordinates and various parameters for time-lapse acquisition. A custom fusion program was used for visualization and generation of maximum intensity projections[42].

For long-term time-lapse acquisition, embryos were embedded in 0.1% low-melting-point agarose inside fluorinated ethylene propylene (FEP) tubes as described in[43–45]. 0.016% tricaine was used in the E3-filled-imaging chamber only when α-bungarotoxin mRNA was not co-injected with DNA constructs.

**Immunostaining and imaging of fixed embryos.** Embryos were immunostained as described previously[16]. Briefly, dechorionated and anesthetized embryos were fixed in 4% paraformaldehyde, 4% sucrose, and 0.1% Triton X-100 in PBS overnight at 4 °C, blocked with 10% goat serum for 1 h, then incubated overnight at 4 °C using any of the following antibodies: chicken α-GFP (Abcam, 1:2000), rabbit α-p63 (GeneTex, 1:100), mouse α-E-cadherin (BD Biosciences, 1:200), rabbit α-caspase-3 (BD Biosciences, 1:100), mouse α-N-cadherin (BD Biosciences, 1:100), mouse α-ZO-1 (Thermo Fisher Scientific, 1:50), mouse α-acetylated tubulin (Sigma-Aldrich, 1:100), rabbit α-L-plastin (gift from Michael Redd [43], 1:400), all of which are described in Supplementary Table 1. After six 20-minute washes with 0.5% PBST (PBS + 0.5% Triton-X), embryos were incubated with appropriate secondary antibodies (goat α-chicken-AlexaFluor-488, goat α-rabbit-AlexaFluor-568, or goat α-mouse-AlexaFluor-647 (Thermo Fisher Scientific, at 1:200) in 10% goat serum overnight at 4 °C, followed by four 20-minute washes with 0.5% PBST. Nuclei were stained with 1 μM DAPI (4′,6-diamidino-2-phenylindole) for 30 min and washed twice with 0.5% PBST. The immunostained embryos were taken through a glycerol series (25%, 50%, 70% glycerol in PBS), mounted in 70% glycerol in PBS or ProLong Gold (Thermo Fisher Scientific) with a #1.5 glass coverslip, and imaged using a Nikon A1R galvano scanning system (20X Plan Apo objective), Leica SP8 white light laser point-scanning confocal microscope (20X Plan Apo CS2 0.75 objective), Zeiss LSM 880 Airyscan microscope (LD LCI Plan-Apochromat 25×0.8 objective), or a Yokogawa CSU-W1/Nikon Eclipse Ti2 spinning disk system equipped with an Andor iXon EMCCD camera.

**Drug treatments.** Rockout (50 mM), chloroquine (50 mM), dynasore (30 mM), and dextran-AlexaFluor-568 (10000 MW, 50 mg/mL) were dissolved in DMSO at the indicated concentrations, aliquoted, and stored at −20 °C. Working solutions in PTU-E3 are indicated in figures. *Rockout and chloroquine:* 24-hpf injected embryos were distributed into 6-well (30 embryos/well) or 24-well (10 embryos/well), E3 aspirated and quickly replaced with drug (or DMSO control) in PTU-E3 for 24 h (48 hpf), then washed, anesthetized with tricaine and fixed, and immunostained as described above. *Dynasore:* Injected 50%-epiboly or shield embryos in 6-well plates at 30 embryos/well) were incubated with dynasore (or DMSO control) in PTU-E3. At 24 hpf, embryos were dechorionated, then incubated in dynasore and 2 mg/mL dextran-AlexaFluor-568 in the dark for 6 h. Embryos were rinsed 3X in E3, incubated in fresh PTU-E3 without drug or tracer in the dark for 2.5 h, anesthetized with tricaine, fixed, and immunostained, as described above.

**In situ mRNA hybridization and immunohistochemistry.** In situ mRNA hybridization for *snai1b*, previously described[43], was done as follows: 2 dpf PTU-E3-treated embryos fixed in 4% paraformaldehyde in PBS overnight at 4 °C, were treated with 50 μg/mL proteinase K (Sigma-Aldrich) for 17 min and hybridised with a digoxigenin (DIG)-tagged antisense *snai1b* probe[44] at 60 °C overnight. After in situ hybridization, embryos were incubated with an α-digoxigenin Fab fragment directly conjugated to alkaline phosphatase (Roche, 1:5000) overnight at 4 °C and developed colorimetrically using NBT/BCIP solution (Roche). Finally, embryos

were inmmunostained for GFP and E-cadherin and DNA (DAPI), as described above.

Embryos were mounted in ProLong Gold (Molecular Probes) with a #1.5 glass coverslip and imaged using a Nikon Eclipse Ni-E FN Upright. *snai1b*-positive cells were identified and imaged using a colour camera and subsequently fluorescence and transmitted light images were acquired with the same pixel size to check for signal colocalization.

**Quantification and statistical analysis**

*Quantifications.* Cell masses: 1 dpf embryos were anesthetized in 0.02% tricaine and dechorionated to count cell masses manually using a brightfield dissection microscope. *Live extrusions:* Live cell extrusions were quantified from time-lapse movies and categorized as apical or basal using orthogonal slice projections. *Internalized vs. surface cells:* Internalized and surface cells were counted from confocal images in orthogonal slice projections, immunostained for p63 or E-cadherin to define the basal cell layer or the periderm, respectively, of the zebrafish epidermis. Cells were classified as live or dead using caspase-3 immunostaining and cell morphology in control- and drug-treated embryos. Cells misexpressing GFP (in muscle, notochord, melanocytes), an artifact of F0 transgenics, were recorded but excluded from invaded cell scoring and statistical analyses. *Circulating tumour cells:* CTCs were quantified in live, dechorionated embryos using a dissection microscope. *Mapping of cell masses:* The number and location of masses were counted using live, dechorionated embryos using a dissection microscope for cell masses, using a microinjection tip to reposition embryos to observe both sides. *Mapping of live extrusions:* The numbers and locations of apical and basal extrusions were mapped from movies of injected Et(periderm:Gal4; UAS:Lifeact-mCherry) embryos. For Supplementary Figs. 1 B–D, notochord, muscle, and melanocyte cells misexpressing GFP from Et(periderm:Gal4 wild-type embryos were counted and compared to the number of cells scored as invaded cells in our assay.

*Ectopic expression of snai1b:* To quantify if injection of embryos with EGFP-CAAX or EGFP-T2A-KRas[V12] affects *snai1b* expression, each embryo was imaged from both lateral sides using a colour camera. Then images were randomized and, using the endogenous expression of *snai1b* in uninjected embryos as a guideline, three independent people assed the expression of the transcription factor without knowing what was the construct injected in each embryo. If the expression of *snai1b* deviated from that of uninjected embryos, i.e., by the appearance of purple dots resembling to individual cells, embryos were quantified to have ectopic *snai1b* expression. In case of discrepancy, the embryo was quantified according to the criterion of the majority.

*Statistical analyses.* Statistics in all graphs were calculated using either Graphpad prism 9.0 or Matlab R2019b, both of which compute p values as an upper limit, rather than an exact number. All tests were run as two-sided tests. To evaluate sample means, we used a non-parametric Mann–Whitney test[46]. To compare more than two groups, we used a Kruskal–Wallis test to reject the null hypothesis, and a Mann–Whitney test with the Holm-Sidak adjustment for pairwise comparisons. To analyze paired data, we used a Wilcoxon signed-rank test. For comparisons of more than two groups of paired data we used the Friedman test to reject the null hypothesis and a Wilcoxon signed-rank test with a Holm-Sidak correction for multiple pairwise comparisons.

For categorical data we used the Chi-Square test to reject the null hypothesis with the Yates correction when analyzing two populations and two categories and with the Holm-Sidak adjustment for pairwise comparisons when comparing more than two groups. For analysis of categorical data, when at least 20% of the groups presented frequencies lower than 5 for a given variable, we used the Fisher Exact test.

*Statistics and reproducibility.* Fig. 4a, b, c', and d represent >30 embryos, whereas e is from 7 embryos, quantified in Supplementary Fig. 5d. Supplementary Fig. 1a represents >30 embryos. d represents 17 wild-type embryos and 14 periderm:Gal4 enhancer trap embryos, quantified in Supplementary Fig. 1b. Supplementary Fig. 2b is representative of 15 embryos, and c of 5 embryos. Supplementary Fig. 4 a represents >30 embryos and b of 18 embryos. Supplementary Fig. 5e is from 7 embryos, quantified in Fig. 4f, f and g represent >30 embryos, and h of 5 or 8 embryos treated with vehicle control or Rockout, respectively, quantified in Fig. 4j.

**Reporting summary.** Further information on research design is available in the Nature Research Reporting Summary linked to this article.

**Data availability**
The raw data that support the findings of this study are available from the corresponding author upon reasonable request. All the other data are available within the article and its Supplementary Information. Source data are provided with this paper.

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

## Acknowledgements

We thank Adam Gardner for early studies that were not included in this report; Channing Der, George Eisenhoffer, Conan Kinsey, and David Grunwald for plasmids to construct KRas and cMyc expression vectors; Kristen Kwan for Tol2kit reagents and help; Rodney Stewart for the p53mut zebrafish line and snai1b probe plasmid; and Masa Tada for the ZO-1 antibody. We thank Simon Hughes, Claudia Linker and Vikki Williams-Ward for technical help and reagents for in situ hybridization, Russell Bell for assistance in graphing Fig. 2C, and Minna Roh-Johnson and James Gagnon for helpful feedback on our manuscript. A National Institutes of Health R01GM102169 and a Howard Hughes Faculty Scholar Award 55108560 to J.R., EMBO Long-Term Fellowships to T.Z.C. ALTF 1130-2018 and G.M.S. ALTF 1078-2015 and P30 CA042014 awarded to Huntsman Cancer Institute core facilities supported this work. We thank the University of Utah zebrafish and fluorescence microscopy cores, the King's College London zebrafish facility and Nikon Imaging Centre, and the Max Planck Institute of Molecular Cell Biology and Genetics Light Fluorescent Microscopy Facility. An NCRR Shared Equipment Grant #1S10RR024761-01 paid for microscopy equipment.

## Author contributions

J.R., J.F. and T.Z.C. designed experiments, interpreted, analyzed data, and co-wrote the manuscript. J.F. made zebrafish Tol2 constructs and performed most live and fixed imaging experiments throughout the paper. T.Z.C. performed the in situ hybridization experiments and quantifications, all the statistical analyses. G.M.S. developed the first fish expressing KRas and CAAX constructs and performed the initial experiments showing that KRas cells extrude basally in zebrafish and get into the bloodstream, designed experiments, interpreted data, and performed all light sheet microscopy. N.M.R. and M.F.J. maintained zebrafish lines, injected and immunostained embryos, and quantified many of the experiments. S.D. and M.J.R. operated mSPIM and wrote code for image acquisition and visualization. J.H. supported G.M.S. and S.D. for SPIM imaging. D.H. mapped the location of cell masses and extrusions, performed the cMyc experiments, and helped genotype p53 mutants.

## Competing interests

The authors declare no competing interests.
