## [Peer Review File · Nature Communications]

REVIEWER COMMENTS

Reviewer #1 (Remarks to the Author):

In this manuscript, Fadul and colleagues follow up on previous work on epithelial carcinogenesis from the Rosenblatt lab carried out in cell culture systems. These former studies had revealed that basal cell extrusion – as a potential first step of metastasis - occurs in epithelial cells in the presence of mutated KRAS (KRASV12) by suppressing alternative apical cell extrusions, which would have tumor-suppressive effects. In the current manuscript, the authors validate and extend these findings, using zebrafish embryos as an in vivo system. Applying state-of-the art in vivo imaging, they show that KRASV12-expressing peridermal cells not only basally extrude, but invade into deeper tissues of the organism and adopt characteristics of mesenchymal cells (N-cadherin or *snai1b*) or neuronal cells (morphological changes and acetylated tubulin). These effects are specific to KRASV12-transformed peridermal cells, as expressing cMyc in a similar manner has no effect on cell extrusions or invasion. Additionally, the number of invaded cells increases drastically in the presence of mutated p53, indicating that p53-mediated cell death is likely an obstacle to the seeding of distant tumors by invaded KRASV12 cells. Interestingly, basal extrusions also occur in regions without peridermal aggregates, thus independently of KRASV12-mediated peridermal hyperplasia, and even, although at much lower rates, in wild-type embryos. Finally, concomitant with basal extrusion of KRASV12 peridermal cells, apical membrane-bound determinants of epithelial cell polarity are pinched off, pointing to a blebbing mechanism which might contribute to the acquisition of new (mesenchymal) cellular characteristics and identities of invading epithelial KRASV12 cells .

This manuscript makes several novel and important points that should be of high interest to the broad scientific audience of Nature Communications, namely (1) the establishment of a vertebrate model in which metastasis can be visualised and followed in real time, including the transport of metastasizing cells via the bloodstream, (2) that basal extrusion / metastasis can occur independently of cellular hyperproliferation, but requires the suppression of apoptotic processes to allow the survival of metastasizing cells as a prerequisite for the seeding of distant tumors, and (3) that basally extruded cells undergo apical membrane blebbing during which they get rid of apical epithelial cell polarity determinants, which might allow them to adopt new cellular characteristics. Although these are novel and very interesting findings, we believe that there are alternative explanations for much of the observations regarding the fate of invading cells following BCE, which are not experimentally addressed in the manuscript. Therefore, to rule out these alternative explanations and to strengthen the authors' conclusions, we propose both major and minor changes, including additional experiments, as outlined below.

Major concerns

- The authors quantify internal GFP-labelled cells, including “cells in circulation,” “neuron-like cells,” and “N-cadherin+ cells,” and compare numbers in the GFP-CAAX labelled negative control construct

to the GFP-KRASV12 construct. A direct comparison of the two conditions relies on the assumption that both constructs behave similarly apart from the KRASV12 expression. However transient expression in embryos driven by plasmid injection frequently gives rise to ectopic expression of the construct. It is impossible to control where in the genome a Tol2-mediated integration will occur and often results in unpredictable expression patterns. Importantly, this very often includes expression in neurons and other cell types including muscle and immune cells. Moreover, any such randomly/ectopically labeled cell would then be more likely to survive and proliferate when it is additionally expressing KRASV12 vs. when it is expressing GFP-CAAX alone. Therefore, the observed GFP-KRASV12 expression in “invaded” or “transformed” keratinocytes could merely represent ectopic expression of the construct in cells which were never peridermal to begin with, and could also explain why, in experiments where *krt4*:GFP-KRASV12 was injected into *krt4*:LifeAct-mCherry embryos, the “invaded” cells express GFP yet seem to have lost mCherry expression (Figure S3 A).

To rule out this alternative explanation, conclusive evidence needs to be shown that invaded cells have derived from peridermal cells. If this is established as part of Figure 2, then the quantification based on transient injection experiments (Figure 3) can remain as shown.

Ideally, this should be shown by a lineage tracing approach using a *krt4*:Cre line with beta-actin:loxPstoploxP-eGFP-KRASV12 or with beta-actin:loxPstoploxP-eGFP-CAAX. Alternatively, injecting an UAS: EGFP-KRASV12 construct into embryos from a cross of heterozygous ET(periderm:Gal4) parents, followed by comparing internal cells in ET(periderm:Gal4) + and ET(periderm:Gal4)- embryos. The minimal requirement would be to inject the plasmid into the *krt4*:mCherry stable line and prove that all internalized cells are actually green and red.

- Figure 3A and movie S7 show a basally extruding cell that then migrates throughout the fish, but in the images it is not totally clear that the moving cell was the KRASV12-expressing cell as the GFP is no longer visible. Does this cell lose GFP for some reason? Was this also the line labelling GFP only in the apical membrane? If so, please make this clear. Alternatively, the movements of the invaded cell could also indicate Lifeact-mCherry labelled cell remnants engulfed by an immune cell immediately following BCE. To exclude this possibility, please inject the *krt4*:eGFP-KRASV12 construct into a transgenic line labeling immune cells (e.g. *lyz*:mCherry) to see if immune cells are found beneath BCE sites and if they consume BCE cells. Additionally, it could be very helpful using a transgene labelling nuclei (as the *h2afva*:*h2afva*-mCherry reporter line used in other experiments in this manuscript) to actually follow the nucleus of the extruded cell. This approach would clarify that the cell survives.

- In Figure S1 B-C, *krt4*:mCherry-cMyc was used as a negative control to show that “merely upregulating growth and proliferation is not sufficient to drive BCE or invasion” (page 4). Yet, it is not shown that proliferation is affected by transient expression of this construct. Please demonstrate increased proliferation rates, using BrdU incorporation, phospho-histone H3 antibody staining, or simply by counting the numbers of dividing cMyc+ and cMyc- peridermal cells in the movies.

- The hypothesis that basally extruded cells change their character by mechanical stripping of apical and epithelial determinants is novel and intriguing. However, it is difficult both to visualise this with the images provided, and also to comprehend technically why E-cadherin, which localizes basolaterally in peridermal cells (see for example Arora et al 2020 Elife), should be lost by this mechanism. This point could be made much more strongly with more convincing images.

- Figure 4G: It is difficult to understand the orientation of this image because there are a lot of purple puncta above the E-cadherin constriction. Where is the surface of the epidermis? Instead of an XZ-projection, a cryosection with anti-GFP and anti-E-cadherin antibodies plus DAPI to label nuclei would make the point much clearer. Additionally, a TEM image showing the ultrastructure including the intact nucleus of the invaded cell and the remnants of the apical membrane would be most conclusive. Can you rule out that the E-cadherin constriction indicated by the arrowhead in this image is from the peridermal cells neighbouring the invading cell? To rule this out, the E-cadherin-YFP or E-cadherin-RFP plasmid (see for instance Bosze B et al., Histochem Cell Biol. 2020) could be injected to show E-cadherin localization in individual cells.

- In the text discussing E-cadherin in the context of cell identity, E-cad is sometimes referred to as an “apical epithelial marker” (page 7). Please revise the text to read “epithelial marker” only, when referring to E-cadherin. Clearly, at the shown stages, E-cadherin does not show a polar distribution, but is found in all membrane domains of peridermal cells! Please also amend the model in Figure S5, where E-cadherin appears to be in tight junctions, instead of in adherens junctions.

Minor concerns

- The Materials and Methods are rather incomplete and should be revised. For example, how was the *krt4:mCherry-cMyc* construct made? How much of it was injected? Which p53 mutant was used? Which p53 MO (give the sequence)?

- Some of the figures are labeled somewhat confusingly. For example, Figure 2E: Blue square should be “not integrated” or “surface” instead of “none”; Figure 4H: the Y axis should read “% of embryos with GFP+ cells” since the “epidermal” category indicates non-invaded cells.

- Please clearly indicate the age and the genotype of the fish in all figure legends (Fig 2 is 5 dpf, Fig 3 is 2 dpf etc.), and clearly indicate which constructs are used (EGFP-KRas or EGFP-T2A-KRas).

- Figure 1. Please indicate how you determine what constitutes a mass of cells and what doesn't. Is it similar to Figure S4, where 2-3 cells in close proximity is defined as a mass?

- Figure 1D. The difference between apical and basal extrusion is difficult to distinguish for an untrained eye in the maximum intensity projection, and the virtual XZ reconstruction does not clearly show a basally extruding cell. This could be complemented with a supplementary movie displaying the z-stack. Additionally, it would be helpful to use e.g. H2A-mCherry to visualize the nucleus. Finally, in the figure legend, it is stated that constricting F-actin is shown by the periderm:Lifeact-EGFP, yet GFP is also present in the injected constructs and labels membranes. It is not possible to conclude that the constriction is indicating F-actin and not membrane compaction. Please adjust the text to remove the reference to F-actin constriction, or show similar images with the constructs injected into the periderm:Lifeact-mCherry instead.

- Figure 2B quantifies “dead,” “internal,” and “surface” cells, but it is not indicated how “dead cells” were defined and quantified. Please include an example image of activated Caspase 3 staining together with krt4:GFP in an invaded cell.

- Figure 3 I shows N-cad+ cell counts in the KRASV12-injected fish only. Were there 0 N-cad+ cells per embryo with GFP-CAAX injection (if so, include in graph), or was this not examined? Does the graph indicate N-cadherin+/GFP+ double positive cells? If so, shouldn't the Y axis read % N-cadherin+/GFP+ / GFP+ cells? For 3H, it's not clear that what is indicated is actually cellular- please include nuclear staining. Additionally, N-cadherin should not be cytoplasmic yet the staining seems to be throughout the entire cell, can you either show a positive control of a cell normally expressing N-cadherin in a wild type fish, or can you comment on the antibody staining? Can you comment whether N-Cadherin+/GFP+ cells are found in specific tissues?

- Figure S3A: Does the neuron also express Life-Act? A close up and showing single channels could show that, which would also strengthen the hypothesis that the cell derived from a peridermal cell.

- Figure S3B: the text indicates that arrowheads show co-localization, but the arrowheads are missing from the figure.

- Figure S3C: *snai1b* in situ- please define “ectopic” expression. Snail is expressed as part of normal embryonic development, please indicate in the text and also in the figure if you also see expression in the expected tissues. If so, where?

- Figure 4E: It is not clear what exactly was quantified for the counts shown in the graph. How was *snai1b* detected? Is this quantification from the images shown in S4E, are all of these cells double positive for *snai1b* and EGFP? If so, what is the percentage of double-positive cells compared to all EGFP-positive cells? What does it look like in CAAX-injected embryos?

- Figure 4F, Movie S13. Whereas it is obvious that lots of apical membrane is pinched off, it is not clearly visible that the cell is basally extruding as the Lifeact-mCherry from this cell cannot be

followed. Here again, a nuclear marker might be useful and the images should be complemented with orthogonal views. How do the authors determine that the red cell appearing from below (arrow at 1:50) is actually the cell that had extruded before?

- Figure S4A: is the green smear towards the bottom of the image an invaded cell? Please add an arrowhead indicating the invaded cell.

- Figure S4B. In this graph, the percentage of invaded cells per embryo negative and positive for ZO-1 is shown. If one embryo contains 90% negative cells, it obviously has 10% positive cells, so it is not necessary to show both. Also, to me, a statistic test doesn't make sense in that case. Please amend this graph.

- Figure S4F. How is actin stained? Is this also the krt4:Lifeact-mCherry stable line? If so, please indicate. Why are invaded cells negative for actin?

- The labeling in the figures and the figure legends of the membrane-tethered EGFP-KRAS vs the 2A-KRAS is inconsistent (the membrane EGFP-KRAS is referred to as EGFP-dt-KRAS only once, but seems to be the same as EGFP-KRAS). Please indicate in the figures and figure legends when the 2A-KRAS is used, and when the normally localized EGFP-KRAS is used. Furthermore, if the membrane EGFP-KRAS is used in e.g. Figure S1 A, why is the internal cell still GFP+? Shouldn't it mechanically lose the membrane-localized EGFP? Conversely, in Figure 3 A, when the 2A-KRAS (cytoplasmic GFP) is indicated, shouldn't the internal cell keep its GFP? We don't see GFP in the indicated circulating cell, only mCherry. Please clarify as this doesn't seem to follow your hypothesis as we understood it. This also leads to a question that should be elaborated in the discussion: if, according to the hypothesis, invading KRAS cells pinch off the membrane which contains the KRAS protein, yet keep an intact nucleus during and after invasion, why don't they then re-express EGFP-KRAS? If they no longer have membrane-bound KRAS, why are they still transformed when they have eliminated the driver of transformation?

Reviewer #2 (Remarks to the Author):

In this manuscript, John Fadul and colleagues describe a new mechanism of tissue invasion based on atypical extrusion of Ras active cells. By combining long term live imaging in vivo (using zebrafish) and pharmacological/genetic perturbations, they show that mosaic Ras activation (KrasV12) in the epidermis combined with p53 down regulation lead to aberrant basal extrusion followed by tissue invasion and plastic behavior (acquisition of neural-like phenotype and mesenchymal phenotype).

Previous works, including work from the same group, have already characterized the mechanism of expulsion of RasV12 cells from epithelial layer. However, the link between atypical basal extrusion and invasive capacity was not yet clearly demonstrated, especially in vivo. Moreover, the manuscript also describes a number of surprising observations, including severing of cell apical domain, and potential new fate acquisitions by invasive cells.

While these observations are all novel and very interesting, the manuscript suffer from some weaknesses. At this stage, the organization and the data cover a wide range interesting behaviors, but they are a bit hard to connect together and still described rather superficially. For instance, it would be interesting to better document the atypical extrusion and the “severing” of the apical domain of cells. Also, the link between these basal extrusion and the migratory cells is based on rather limited data at this stage.

Overall, a reorganization of the manuscript (maybe by describing these events in their putative chronological order) and few additional movies/quantification (see below for suggestions) would really strengthen these observations (which I believe have great potential).

Major points/suggestions :

1. Previous works, including works in zebrafish, have rather described apical extrusion of RasV12 cells (see for instance work from Y Fujita lab). While I do understand that basal extrusion may have been overlooked in these other studies, could the authors speculate on what could explain the dominant role of basal extrusion in their system (especially compared to Takeushi et al. Curr Biol 2020 where the genetic system seems to be exactly the same) ? It would be fair to have some quick mention of this in the discussion.

2. Many data/movies are rather hard to interpret and it is sometimes hard to connect the interpretation/description in the manuscript with the pictures/movies. Could the author first show systematically separated channels both in figures and movies ? (especially for Fig 4A to G and movie S7, movie S13). For instance, this is very difficult (if not impossible) to see the apical constriction on the lateral view provided in Fig. 4F (where is the E-cad belt of the neighbouring cells ? Where is the constricted apical side ?)

3. The connections between basal extrusion, cell migration and tissue invasion remain quite indirect at this stage. So far, this is mostly based on one movie which show one event of extrusion and some basal body moving around and the link between perturbations reducing basal extrusion and the proportion of ectopic cells in the embryo.

First, could the authors exclude that the basal body shown in Fig. 3A and movie S7 is not an apoptotic fragment transported in some macrophage ? In fact, there are quite many magenta positive particles moving around on the basal side. If possible, it might be usefull to check that there

is no nuclear fragmentation (as one would expect following apoptosis) and that those basal moving bodies do contain the Ras cell nucleus.

Similarly, in Fig. 4F, it is not clear how the authors can connect without ambiguity the Ras extruding cell with the magenta positive particle on the basal side (shown on the last panel, bottom right). Documented tracking in 4D would help here.

Finally, it would be really useful if the authors have captured unambiguously events connecting basal cell extrusion, cell migration, cell mitosis, access to the blood flow and/or abnormal differentiation in single movies. I do appreciate that it might be rare events difficult to catch and track on single embryos, but without that, it will be difficult to clearly connect these different cell populations unambiguously.

4. The severing of the apical domain and the basal extrusion are quite interesting. On the first sight, the multiple blebbing and cell bodies may look like apoptotic fragments. As indicated above, It would be really useful to provide movies of these basal extrusion with a marker of nuclei to show that the main cell body is basal and to exclude any nuclear fragmentation. This would also help to define the localisation of the cell among all these remaining debris (see above)

5. The title of some graphs and the associated legends are quite often ambiguous and it is not always clear what is exactly represented (especially for graph showing percent of something). This includes:

Fig 3G : is the quantification in p53 background ?

Fig 3I: What represents the percent of cells is not so clear, is it the percent of all GFP positive cells expressing N-cad or only the cell internalized (with one dot = one embryo) ? Is this percentage different for the internalized Ras cells and the Ras cells in the epidermis ?

Figure 4E: Is it the percentage of embryos showing snail positive cells in the different localisation or is that the proportion of Snail positive cells for the full Ras population of a given compartment in one embryo ?

6. Fig 4HI : is the total number of surviving Ras V12 cell similar in the controls in the ROCKOUT ? There seems to be much less cells in ROCKOUT treatment (while one would expect to find more cell in the epidermis in this condition). One alternative explanation for the absence of internalized cell could be that this factor affects cell survival after extrusion (and not the exit from the layer).

Other minor points:

- The font size seems to change in the middle of page 3

- Fig 2D: are these global morphological defects commonly observed upon depletion of p53 and induction of RasV12 in a subset of cells ? Do the deformations always correlate with the presence of Rasv12 cells masses ?
- Fig. 2B: How do the authors identify Ras dying cells unambiguously ? I see that the authors mention cleaved caspase3 staining and morphological features in the methods. It would be usefull to give more details and maybe show a representative example of each category.
- In the legend of Fig. 3E, the number of embryos is missing
- There is a rather large litterature in Drosophila documenting how the combination of RasV12 activation with various mutations lead to invasive behaviours (check for instance numerous works from Tian Xu lab, Yale School of Medicine). Although to my knowledge this was never associated with long term live imaging, it might be relevant to compare the behaviour observed in zebrafish with these works (including similarities and differences).

Reviewer #3 (Remarks to the Author):

KRas-transformed epithelia cells invade and partially dedifferentiate by basal cell extrusion.

Fadul et al.

In this paper, Fadul and colleagues describe the phenomenon of basal cell extrusion (BCE) as a mechanism by which Kras-transformed epithelial cells leave an epithelium and disseminate to distant sites. This work stems from previous work in the lab, which has defined cell extrusion as a process to remove excess or dying cells from epithelial tissues without breaching barrier function. Here, they build on previous work and combine zebrafish in vivo models with live cell imaging to study cell extrusion in real time. They find that as well as triggering the formation of neoplasia, Kras-transformation induces BCE of cells, independent of neoplasia. Interestingly, BCE is associated with a change in cell morphology of the extruded cells, leading to the speculation that dissemination is linked to a change in cell fate. Moreover, the authors show that extruded cells are unable to survive unless they lose functional p53. Kras mutations are key driver mutations in pancreatic and lung cancers, both of which are highly metastatic and of unmet clinical need. In addition, there is increasing evidence that metastasis occurs at very early time points and before overt tumour formation, challenging our understanding of metastasis as a progressive step in tumorigenesis. The findings in this paper are novel and important as they support this current view and shed light on the mechanisms underlying early metastasis. While this paper is very interesting and compelling, it requires revision before being considered for publication at Nature Communications. I hope my comments will help to shape discussion and interpretation and improve clarity, particularly for the non-expert.

The main weakness of the paper is that it is not clear whether there is a distinct molecular mechanism(s) underlying how KrasV12 cells decide whether to be extruded apically or basally or overgrow in cell masses, or whether each phenomenon is stochastic. The authors demonstrate a number of mechanisms are required; however, as written it is not clear how each mechanism works together in the system. Is BCE stochastic or specific to a region of the tissue? How does lack of functional p53 contribute to the change in Kras cell fate/morphology or is it only required to increase survival? What is the requirement of cell proliferation/division in the system? Some clarity/discussion on each of these points would strengthen the message.

Minor points to address:

1. Fig 1D illustrates apical and basal extrusion – judging from the IF images, the process is similar in both and is only distinct by the direction of extrusion. Have the authors confirmed direction of extrusion using defined apical and basal markers? Here the authors are tracing GFP yet describe contracting F-actin rings. Do they have data staining for F-actin or similar to support this conclusion? Later they show that inhibition of autophagy inhibits BCE (using chloroquine) Under these conditions, are KrasV12 cells apically extruded instead? This would also help understand mechanism.

2. Experiments are based on mosaic expression of eGFP-KrasV12; however, it is not clear what mosaic expression means in terms of neighbourhood. Can they estimate how much of the tissue is expressing the transgene? Does BCE (and apical extrusion) occur from regions of the epithelium that is predominately wild-type (i.e., where mutant cells have normal neighbours) or are labelled cells being extruded from clones/clusters of transformed cells?

3. Fig. 1B describes cell masses per embryo and relates to main text “masses of 3-40 KrasV12 cells formed...” what is the timeline here? What time post fertilisation does Fig. 1B relate to? A definition of ‘hpf’ in the figure legends would help the non-expert. The maps in Fig. 1C and 1F suggest that cell masses form in early embryos; whereas extrusion occurs later in development. Please comment.

4. The authors describe how cells expressing c-Myc did not form masses, are not extruded or internalised (top of Page 4). This is assuming that KrasV12 cells are internalised; however, this appears very suddenly, and it is not clear which of the data describes internalised Kras cells. My interpretation is that KrasV12 cells invade the basal epidermal layer; yet as written the authors suggest that transformed cells are encapsulated in other cells. Please clarify with supporting evidence. If it is indeed that extruded cells invade the basal layer, I would argue that ‘internalised’ is misleading here and needs to be reconsidered as a description in subsequent text/figures.

5. The authors conclude that overexpression of oncogenic c-Myc upregulates growth and proliferation and this has no effect on BCE, transformed cell mass or invasion. Do the authors have evidence to show with confidence that c-Myc overexpression does indeed increase growth/proliferation in vivo? Overexpression of PI3K signalling may be a better control here. This is

important because the data demonstrate that extrusion occurs at sites of cell division; therefore, increasing cell division rates may increase extrusion and provide a possible molecular mechanism for extrusion.

6. The authors claim that transformed cells that are basally extruded and survive, “adopt new plasticity”. The authors use acetylated tubulin and N-cadherin to define the new cell states, concluding that the cells adopt neuronal/mesenchymal fate. My interpretation is that the authors chose these markers based on changes in cell morphology rather than plasticity. The term ‘plasticity’ suggests a reprogramming to a stem-like fate. Have the authors screened for additional markers in these cells to conclude that the cells adopt a plastic state; e.g., markers of stemness or EMT markers. E.g., Zeb-1 has been shown to be a marker of early circulating tumour cells in pancreas. Indeed, they show that cells enter the circulation; are circulating cells also stem-like?

7. Top of page 7 – the authors conclude that BCE cells lose epithelial markers but retain mesenchymal markers. This is based on the fact that expression of eGFP is from the keratin 4 promoter (not clear why they refer to Fig. 2E to support this). In my view, an additional (EMT) marker is required to show mesenchymal cell status and support this conclusion.

8. Mechanistically, the authors show that loss of E-cadherin is not likely to be dependent on changes in transcription or dynamin-mediated endocytosis. They present compelling data to show that E-cadherin is lost via pinching of an actin ring during extrusion process. Inhibition of Rho kinase (ROCK) also blocks BCE and invasion. While the authors claim that this is distinct from extrusion of v-Src-expressing cells (Anton et al., 2018), do the authors have sufficient evidence to support this claim; is myosin activity also required in the actin ring formation and pinching?

9. The schematic in Fig S5 is misleading as cancer cell invasion occurs from a cluster of transformed cells that are also apically extruding. In the text (final paragraph page 10) the authors claim that BCE occurs independent of cell masses, which is not illustrated in the cartoon. This comes back to my first point (where in tissues are KrasV12 cells being basally eliminated from) and needs clarification.

Thanks very much for reviewing our manuscript. We have been able to provide the additional data, in new supplemental data and movies and have changed the text, which we mark in blue now. We respond to individual comments below in blue:

Reviewer #1

In this manuscript, Fadul and colleagues follow up on previous work on epithelial carcinogenesis from the Rosenblatt lab carried out in cell culture systems. These former studies had revealed that basal cell extrusion – as a potential first step of metastasis - occurs in epithelial cells in the presence of mutated KRAS (KRASV12) by suppressing alternative apical cell extrusions, which would have tumor-suppressive effects. In the current manuscript, the authors validate and extend these findings, using zebrafish embryos as an *in vivo* system. Applying state-of-the art *in vivo* imaging, they show that KRASV12-expressing peridermal cells not only basally extrude, but invade into deeper tissues of the organism and adopt characteristics of mesenchymal cells (N-cadherin or *snai1b*) or neuronal cells (morphological changes and acetylated tubulin). These effects are specific to KRASV12-transformed peridermal cells, as expressing cMyc in a similar manner has no effect on cell extrusions or invasion. Additionally, the number of invaded cells increases drastically in the presence of mutated p53, indicating that p53-mediated cell death is likely an obstacle to the seeding of distant tumors by invaded KRASV12 cells. Interestingly, basal extrusions also occur in regions without peridermal aggregates, thus independently of KRASV12-mediated peridermal hyperplasia, and even, although at much lower rates, in wild-type embryos. Finally, concomitant with basal extrusion of KRASV12 peridermal cells, apical membrane-bound determinants of epithelial cell polarity are pinched off, pointing to a blebbing mechanism which might contribute to the acquisition of new (mesenchymal) cellular characteristics and identities of invading epithelial KRASV12 cells.

This manuscript makes several novel and important points that should be of high interest to the broad scientific audience of Nature Communications, namely (1) the establishment of a vertebrate model in which metastasis can be visualised and followed in real time, including the transport of metastasizing cells via the bloodstream, (2) that basal extrusion / metastasis can occur independently of cellular hyperproliferation, but requires the suppression of apoptotic processes to allow the survival of metastasizing cells as a prerequisite for the seeding of distant tumors, and (3) that basally extruded cells undergo apical membrane blebbing during which they get rid of apical epithelial cell polarity determinants, which might allow them to adopt new cellular characteristics. Although these are novel and very interesting findings, we believe that there are alternative explanations for much of the observations regarding the fate of invading cells following BCE, which are not experimentally addressed in the manuscript. Therefore, to rule out these alternative explanations and to strengthen the authors' conclusions, we propose both major and minor changes, including additional experiments, as outlined below.

Major concerns

- The authors quantify internal GFP-labelled cells, including “cells in circulation,” “neuron-like cells,” and “N-cadherin+ cells,” and compare numbers in the GFP-CAAX labelled negative control construct to the GFP-KRASV12 construct. A direct comparison of the two conditions relies on the assumption that both constructs behave similarly apart from the KRASV12 expression. However transient expression in embryos driven by plasmid injection frequently gives rise to ectopic expression of the construct. It is impossible to control where in the genome a Tol2-mediated integration will occur and often results in unpredictable expression patterns. Importantly, this very often includes expression in neurons and other cell types including muscle and immune cells. Moreover, any such randomly/ ectopically labeled cell would then be more likely to survive and proliferate when it is additionally expressing KRASV12 vs. when it is expressing GFP-CAAX alone. Therefore, the observed GFP-KRASV12 expression in “invaded” or “transformed” keratinocytes could merely represent ectopic expression of the construct in cells which were never peridermal to begin with, and could also explain why, in experiments where *krt4*:GFP-KRASV12 was injected into *krt4*:LifeAct-mCherry embryos, the “invaded” cells express GFP yet seem to have lost mCherry expression (Figure S3 A).

- The ectopic expression of transient transgenics was an initial concern of ours, however, we find that we get the same numbers of mis-expressed cells in both *krt4*:EGFP-CAAX and *krt4*:EGFP-KRas^{V12}. We show this in Figure S1B now.
- The picture in (now) Fig. S4A is in fact from a *krt4*:EGFP-KRas^{V12} plasmid injected into a periderm:Gal4 driver of Lifeact, so we would not necessarily expect continuous expression of the periderm driver, once invaded.

To rule out this alternative explanation, conclusive evidence needs to be shown that invaded cells have derived from peridermal cells. If this is established as part of Figure 2, then the quantification based on transient injection experiments (Figure 3) can remain as shown.

Ideally, this should be shown by a lineage tracing approach using a *krt4*:Cre line with beta-actin:loxPstoploxP-eGFP-HRASV12 or with beta-actin:loxPstoploxP-eGFP-CAAX. Alternatively, injecting an UAS: EGFP-KRasV12 construct into embryos from a cross of heterozygous ET(periderm:Gal4) parents, followed by comparing internal cells in ET(periderm:Gal4) + and ET(periderm:Gal4)- embryos. The minimal requirement would be to inject the plasmid into the *krt4*:mCherry stable line and prove that all internalized cells are actually green and red.

- We thank the reviewer for the concrete suggestion of injecting a UAS:EGFP-KRas^{V12} construct into WT versus periderm:Gal4 embryos. We have been working to build a stable Cre inducible line but are still many months away from completing this. We now include Figure S1B, showing that the number of mis-expressing cells are consistent in all transient transgenic zebrafish. The graph shows that the numbers of embryos mis-expressing (a few) UAS:EGFP-KRas^{V12} cells is similar in a wild-type background versus a periderm:Gal4 background, similar to those in *krt4*: EGFP-KRas versus *krt4*:EGFP-CAAX. Importantly, of these, only one embryo (of 27 UAS-GFP-KRas^{V12}-injected WT embryos) had a single internalized independent GFP-KRas cell, on par with that seen in *krt4*:CAAX-injected embryos (Fig.2B&C). If mis-expression of KRas^{V12} led to the internalized cells we scored, we would predict ~800 cells per 27 embryos, instead of the one possible one we scored. Thus, this control indicates that misexpression of KRas^{V12}

in non-epithelial cells does not lead to high survival and proliferation rates, associated with peridermal invading KRas-transformed cells.

- Figure 3A and movie S7 show a basally extruding cell that then migrates throughout the fish, but in the images it is not totally clear that the moving cell was the KRASV12-expressing cell as the GFP is no longer visible. Does this cell lose GFP for some reason? Was this also the line labelling GFP only in the apical membrane? If so, please make this clear.
 - The cell, which is expressing the non-directly tethered GFP, labeled '2A-KRas in green', retains GFP but the brightness goes down with BCE, likely because a portion of the cell has been lost. Because this is easier to see in red/green (for people who are not red-green colour blind), we now include the Supplemental movie in red/green, keeping the stills in Figure in magenta/green. Throughout the text, we refer to non-directly tethered as EGFP-KRas^{V12} and directly tethered as EGFP-dt-KRas^{V12}. If you can suggest another way to do this, please comment.

Alternatively, the movements of the invaded cell could also indicate Lifeact-mCherry labelled cell remnants engulfed by an immune cell immediately following BCE. To exclude this possibility, please inject the krt4:eGFP-KRASV12 construct into a transgenic line labeling immune cells (e.g. lyz:mCherry) to see if immune cells are found beneath BCE sites and if they consume BCE cells. Additionally, it could be very helpful using a transgene labelling nuclei (as the h2afva:h2afva-mCherry reporter line used in other experiments in this manuscript) to actually follow the nucleus of the extruded cell. This approach would clarify that the cell survives.

- We now include one (of several) movies (Movie S9) of KRas^{V12} cells in an mpeg:mCherry zebrafish line that labels macrophages and included confocal projections of zebrafish immunostained for macrophages with L-plastin (S. Fig. 2C), indicating that there is overlap with either. The finding that macrophages do not engulf live invading cells and we do not see them co-staining for macrophages by immunofluorescence using an antibody to EGFP, which is pH insensitive, shows that the internalized cells we see migrating, dividing, and morphing are not within macrophages.
- Additionally, we also include Fig. S2A&B showing stills from Movie S7 and immunostaining for active caspase-3 indicating the phenotypes seen when a cell invaded by BCE dies. In the stills and the movie expressing a H2B:RFP, you can see that the dying cell remnant gets engulfed, causing rapid disappearance of the EGFP signal, due to quenching by low pH. (This transgene does not allow one to easily follow BCE, however.) This demonstrates that cells invading by BCE no longer fluoresce green following engulfment by macrophages. This engulfment and loss of fluorescence is a standard response to the minority of invaded cells that die is easily distinguished from cells that continue to fluoresce EGFP following invasion by BCE.
- Finally, inclusion of two more movies (16 &17) show that cells pinch off EGFP-dt-KRASV12 but continue to express cytoplasmic NTR-mCherry, as they migrate away. If these cells were consumed by macrophages to migrate, one would see red cells migrate to them, and engulf them prior to migrating, but this does not happen.

- In Figure S1 B-C, krt4:mCherry-cMyc was used as a negative control to show that “merely upregulating growth and proliferation is not sufficient to drive BCE or invasion” (page 4). Yet, it is not shown that proliferation is affected by transient expression of this construct. Please demonstrate increased proliferation rates, using BrdU incorporation, phospho-histone H3 antibody staining, or simply by counting the numbers of dividing cMyc+ and cMyc- peridermal cells in the movies.

- You make a good point. We cannot really say what cMyc might be doing in this case. Because investigating this would deflect from the main point of the paper, we have decided to instead state that oncogenic cMyc or tumor suppressive p53 mutations are not sufficient to cause invasion by BCE. Instead, we state, **“By contrast, cMyc over-expression, implicated in metastasis²⁴, did not cause masses, extrusion, or internalized cells, suggesting that not all oncogenic signalling drives BCE or invasion (Fig. S1C,D & Movie S6).”**

- The hypothesis that basally extruded cells change their character by mechanical stripping of apical and epithelial determinants is novel and intriguing. However, it is difficult both to visualise this with the images provided, and also to comprehend technically why E-cadherin, which localizes basolaterally in peridermal cells (see for example Arora et al 2020 Elife), should be lost by this mechanism. This point could be made much more strongly with more convincing images.

- To address this, we have added to now Fig. S5F, which shows high resolution imaging of E-cadherin within the epidermis. Here, the E-cadherin localizes with and above the actin in the periderm, but basolaterally within the basal epidermal layer. Because the staining in the basal layer is very prominent, this may appear confusing at first glance, so we show several points within the periderm that clearly show it localizes with and above apical cortical actin. The Arora paper demonstrates E-cadherin below APKC but does not include data referencing its location with respect to actin. Since actin and myosin are the key factors here with respect to ring contraction, we think that this the main relevant protein to counterstain for and it clearly indicates that E-cadherin is with and apical to actin, where it can be pinched off with actomyosin constriction.
- Additionally, the one basal extrusion, we managed to fix for immunofluorescence (it is a fast process so hard to catch stochastically), shows that E-cadherin is clearly above the bulk of the cytoplasmic GFP (Fig. 4G).
- Finally, the directly tethered GFP-dt-KRas^{V12} in Movie 15 clearly shows it at the apex of all cells and then pinched off through during BCE. We have included Movies 16 & 17 to also show other examples of GFP-dt-KRas^{V12} getting pinched off during BCE, yet retaining cytoplasmic mCherry fluorescence as they migrate away.
- We now add to page 5, **“This severing of the apical membrane explained why all cells invading by BCE experienced concomitant membrane blebbing and loss of green fluorescence before migrating from sites of extrusion (e.g. Movies S8, 16, & 17 Fig. 3A’). Moreover, E-cadherin colocalizes with and slightly above apically localized actin in other studies³³ and in the periderm and becomes constricted into an apical point above a cell during BCE (Fig. 4G and SFig. 5F), suggesting it is also stripped from the invading cell.”**

-

- Figure 4G: It is difficult to understand the orientation of this image because there are a lot of purple puncta above the E-cadherin constriction. Where is the surface of the epidermis? Instead of an XZ-projection, a cryosection with anti-GFP and anti-E-cadherin antibodies plus DAPI to label nuclei would make the point much clearer. Additionally, a TEM image showing the ultrastructure including the intact nucleus of the invaded cell and the remnants of the apical membrane would be most conclusive. Can you rule out that the E-cadherin constriction indicated by the arrowhead in this image is from the peridermal cells neighbouring the invading cell? To rule this out, the E-cadherin-YFP or E-cadherin-RFP plasmid (see for instance Bosze B et al., *Histochem Cell Biol.* 2020) could be injected to show E-cadherin localization in individual cells.

- Thank you for this suggestion. Unfortunately, we spent many months trying to get the construct to do this to no avail. We contacted Steffen Scholpp, who gave us a construct (Bosze et al., *Histochem Cell Biol* 2020) that had only ~75% identity to fish E-cadherin upon sequencing and BLAST analysis. His lab did not have any other information this construct, even enough to determine how to linearize and in vitro transcribe it. We've wondered why such a construct would not be used frequently in zebrafish work. Maybe it is hard to clone?
- As mentioned above, the chances of stochastically catching a cell mid-extrusion is so small, that this experiment would be impossible to do by TEM.

- In the text discussing E-cadherin in the context of cell identity, E-cad is sometimes referred to as an “apical epithelial marker” (page 7). Please revise the text to read “epithelial marker” only, when referring to E-cadherin. Clearly, at the shown stages, E-cadherin does not show a polar distribution, but is found in all membrane domains of peridermal cells! Please also amend the model in Figure S5, where E-cadherin appears to be in tight junctions, instead of in adherens junctions.

- As mentioned above, we clearly show that E-cadherin lies with and above apical cortical actin, so we think our model is justified (SFig. 5F). E-cadherin may be basolateral to the microridges on the very apex of the cell or to other markers used in the Arora paper, but it is clearly aligned an apical to the cortical actin, which is relevant when considering where apical constriction occurs during BCE. Additionally, our model is supported by the fact that E-cadherin suddenly vanishes from cells that basally extrude. If it were basolateral, as suggested, it would not disappear. This may be true of cells basally extruding from the basal epidermal layer, where E-cadherin does appear basolaterally, but not in the periderm layer. However, we did not investigate BCE from the basal layer in this paper.

Minor concerns

- The Materials and Methods are rather incomplete and should be revised. For example, how was the krt4:mCherry-cMyc construct made? How much of it was injected? Which p53 mutant was used? Which p53 MO (give the sequence)?

- We have now updated the methods with this information. Thanks for pointing out these oversights.

- Some of the figures are labeled somewhat confusingly. For example, Figure 2E: Blue square should be “not integrated” or “surface” instead of “none”; Figure 4H: the Y axis should read “% of embryos with GFP+ cells” since the “epidermal” category indicates non-invaded cells.

Changed to surface in each case.

- Please clearly indicate the age and the genotype of the fish in all figure legends (Fig 2 is 5 dpf, Fig 3 is 2 dpf etc.), and clearly indicate which constructs are used (EGFP-KRas or EGFP-T2A-KRas).

- Now addressed.

- Figure 1. Please indicate how you determine what constitutes a mass of cells and what doesn't. Is it similar to Figure S4, where 2-3 cells in close proximity is defined as a mass?

- The manuscript (page 2) states: "Mosaic krt4-EGFP-KRas^{V12} expression induced masses of 3-40 cells at the zebrafish fin edges..."

- Figure 1D. The difference between apical and basal extrusion is difficult to distinguish for an untrained eye in the maximum intensity projection, and the virtual XZ reconstruction does not clearly show a basally extruding cell. This could be complemented with a supplementary movie displaying the z-stack. Additionally, it would be helpful to use e.g. H2A-mCherry to visualize the nucleus. Finally, in the figure legend, it is stated that constricting F-actin is shown by the periderm:Lifeact-EGFP, yet GFP is also present in the injected constructs and labels membranes. It is not possible to conclude that the constriction is indicating F-actin and not membrane compaction. Please adjust the text to remove the reference to F-actin constriction, or show similar images with the constructs injected into the periderm:Lifeact-mCherry instead.

- We have changed the text to: **‘Strikingly, EGFP-KRas^{V12} cells extruded apically and basally at high rates, as scored by hallmark basal or apical constrictions, respectively’**. Because we get into this in more detail later as it relates to stripping of apical membrane, we would prefer not to go into mechanistic detail here. We just want to relay the idea that cells are extruding apically and basally alone.

- Figure 2B quantifies “dead,” “internal,” and “surface” cells, but it is not indicated how “dead cells” were defined and quantified. Please include an example image of activated Caspase 3 staining together with krt4:GFP in an invaded cell.

- We have now included a picture of caspase-positive and negative GFP-KRas cells in SFig. 2B and included how we score them in live and fixed samples in the main text at the bottom of page 2.

- Figure 3 I shows N-cad+ cell counts in the KRASV12-injected fish only. Were there 0 N-cad+ cells per embryo with GFP-CAAX injection (if so, include in graph), or was this not examined? Does the graph indicate N-cadherin+/GFP+ double positive cells? If so, shouldn't the Y axis read % N-cadherin+/GFP+ / GFP+ cells?

- Because there were hardly any internalized CAAX cells to score, we did not include them and instead explain this in the figure legends.

For 3H, it's not clear that what is indicated is actually cellular- please include nuclear staining. Additionally, N-cadherin should not be cytoplasmic yet the staining seems to be throughout the

entire cell, can you either show a positive control of a cell normally expressing N-cadherin in a wild type fish, or can you comment on the antibody staining? Can you comment whether N-Cadherin+/GFP+ cells are found in specific tissues?

- We and others see N-cadherin in the cytoplasm in some cases. Data from the AbCam shows cytoplasmic localization of N-cadherin, which can occur in some cancers:

<https://www.spandidos-publications.com/10.3892/ol.2018.7751>

And in lung fibrosis EMT:

<https://journals.plos.org/plosone/article?id=10.1371/journal.pone.0007559>

Or 3D culture of prostate cancer cells:

<https://www.mdpi.com/2073-4409/8/2/143/htm>

• Figure S3A: Does the neuron also express Life-Act? A close up and showing single channels could show that, which would also strengthen the hypothesis that the cell derived from a peridermal cell.

- Since we were driving Lifeact from the enhancer trap-periderm Gal4 line and the GFP-KRas from the krt4 promoter, this would not be expected. It is possible that once invaded, the enhancer trap no longer expresses actin or does so at very low levels.

• Figure S3B: the text indicates that arrowheads show co-localization, but the arrowheads are missing from the figure.

- Thanks, these got pulled underneath the image—we fixed it.

• Figure S3C: snai1b in situ- please define “ectopic” expression. Snail is expressed as part of normal embryonic development, please indicate in the text and also in the figure if you also see expression in the expected tissues. If so, where?

- We indicated in the methods that we used the expression of snai1b in CAAX embryos as a guide for normal, endogenous Snail expression. We believe that ‘ectopic’ is the correct word here.

• Figure 4E: It is not clear what exactly was quantified for the counts shown in the graph. How was snai1b detected? Is this quantification from the images shown in S4E, are all of these cells double positive for snai1b and EGFP? If so, what is the percentage of double-positive cells compared to all EGFP-positive cells? What does it look like in CAAX-injected embryos?

- Snai1b was detected by in situ hybridization. As indicated in the methods and shown in Fig. S4C and S5C, to be able to image simultaneously, the hybridization signal and fluorescence, we detected the enzymatic signal with brightfield. We inverted the bright field signal in the images to be able to overlay it with immunofluorescence staining for GFP-KRas (or GFP-CAAX) and E-cadherin and show colocalization. This was challenging but it actually worked well. This fits well the analysis showing that GFP-KRas-injected zebrafish had ectopic expression of Snai1B in regions not associated with wild-type zebrafish injected with GFP-CAAX, as indicated in all the figures.

• Figure 4F, Movie S13. Whereas it is obvious that lots of apical membrane is pinched off, it is not clearly visible that the cell is basally extruding as the Lifeact-mCherry from this cell cannot be followed. Here again, a nuclear marker might be useful and the images should be complemented with orthogonal views. How do the authors determine that the red cell appearing from below (arrow at 1:50) is actually the cell that had extruded before?

- This is a zoomed in movie of this process that we would prefer to keep, as it nicely shows the pinching off of the apical membrane. The remaining red cell goes out of focus and then back into focus, however, so is not as great at showing this point. Instead, we now also include two (of many) movies showing invasion at lower magnification, where you can see pinching off of green with the remaining red cytoplasmic marker (NTR-mCherry) remains in the cells that migrate away (Movies S16 & S17).

• Figure S4A: is the green smear towards the bottom of the image an invaded cell? Please add an arrowhead indicating the invaded cell.

- Added.

• Figure S4B. In this graph, the percentage of invaded cells per embryo negative and positive for ZO-1 is shown. If one embryo contains 90% negative cells, it obviously has 10% positive cells, so it is not necessary to show both. Also, to me, a statistic test doesn't make sense in that case. Please amend this graph.

- Positives are deleted, as you requested.

• Figure S4F. How is actin stained? Is this also the krt4:Lifeact-mCherry stable line? If so, please indicate. Why are invaded cells negative for actin?

- These are stained for actin with phalloidin, as stated in figure legend. They are not actin-negative, just dimmer, and clearly white with overlap.

• The labeling in the figures and the figure legends of the membrane-tethered EGFP-KRAS vs the 2A-KRAS is inconsistent (the membrane EGFP-KRAS is referred to as EGFP-dt-KRAS only once, but seems to be the same as EGFP-KRAS). Please indicate in the figures and figure legends when the 2A-KRAS is used, and when the normally localized EGFP-KRAS is used.

- We do indicate this in all situations where we are doing live imaging now in the figure legends. In the fixed samples, we immunostain for GFP, as fixation degrades live GFP signal.

Furthermore, if the membrane EGFP-KRAS is used in e.g. Figure S1 A, why is the internal cell still GFP+? Shouldn't it mechanically lose the membrane-localized EGFP? Conversely, in Figure 3 A, when the 2A-KRAS (cytoplasmic GFP) is indicated, shouldn't the internal cell keep its GFP? We don't see GFP in the indicated circulating cell, only mCherry. Please clarify as this doesn't seem to follow your hypothesis as we understood it. This also leads to a question that should be elaborated in the discussion: if, according to the hypothesis, invading KRAS cells pinch off the membrane which contains the KRAS protein, yet keep an intact nucleus during and after invasion, why don't they then re-express EGFP-KRAS? If they no longer have membrane-

bound KRAS, why are they still transformed when they have eliminated the driver of transformation?

- We have already mentioned in the discussion the fact that EGFP-KRas expression is clipped off (all or partially, depending on whether it is membrane-tethered or not, respectively) but then becomes re-expressed later. We believe that this indicates that they undergo a partial EMT where surface epithelial markers like E-cadherin and ZO-1 are lost permanently but cytoplasmic markers, like keratin retain expression. Because KRT4 drives KRas in our movies, it is likely that both keratin and KRas are continuously expressed, causing their reappearance. We state this on page 4 of the discussion, saying that, **“While much of EGFP signal within invading KRas^{V12} cells is initially pinched off, its re-expression from the *keratin 4* promoter, suggests that cells invading by BCE continue to express this cytoplasmic epithelial marker (Fig. 2E). Thus, cells invading by BCE lose cell surface epithelial determinants but retain cytoplasmic ones, with only some adopting mesenchymal features. These findings could address why metastatic tumours can express both epithelial and mesenchymal markers^{30,31}”**.

Reviewer #2 (Remarks to the Author):

In this manuscript, John Fadul and colleagues describe a new mechanism of tissue invasion based on atypical extrusion of Ras active cells. By combining long term live imaging in vivo (using zebrafish) and pharmacological/genetic perturbations, they show that mosaic Ras activation (KrasV12) in the epidermis combined with p53 down regulation lead to aberrant basal extrusion followed by tissue invasion and plastic behavior (acquisition of neural-like phenotype and mesenchymal phenotype).

Previous works, including work from the same group, have already characterized the mechanism of expulsion of RasV12 cells from epithelial layer. However, the link between atypical basal extrusion and invasive capacity was not yet clearly demonstrated, especially in vivo. Moreover, the manuscript also describes a number of surprising observations, including severing of cell apical domain, and potential new fate acquisitions by invasive cells.

While these observations are all novel and very interesting, the manuscript suffer from some weaknesses. At this stage, the organization and the data cover a wide range interesting behaviors, but they are a bit hard to connect together and still described rather superficially. For instance, it would be interesting to better document the atypical extrusion and the “severing” of the apical domain of cells. Also, the link between these basal extrusion and the migratory cells is based on rather limited data at this stage.

- Thank you for these comments. We believe that we have documented quite clearly, especially given the constraints of the journal basal extrusion and migration beneath the layer afterwards. However, to include more examples (of which we have hundreds), we have added SMovies 16 & 17 in addition to SMovie 8 & 15, which we included previously.

Overall, a reorganization of the manuscript (maybe by describing these events in their putative chronological order) and few additional movies/quantification (see below for suggestions) would really strengthen these observations (which I believe have great potential).

- Thank you for this suggestion. While we considered and even trialed this way of introducing the paper, it disrupted the introduction of several important concepts that we would like to emphasize.
 - With this layout, it made it difficult to discuss how p53 acts on the survival of cells. This is an important concept, as most cancer studies consider p53 loss a 'later' event in metastasis. However, we believe that it represents a step that allows survival of cells that have invaded.
 - Additionally, for storytelling purposes, we feel that one does not really care about the clipping off of the apical membrane until we identify that all cells lose surface epithelial markers as they invade. At this point, the mechanism becomes more compelling.

Therefore, we prefer to present the story in its current order.

Major points/suggestions :

1. Previous works, including works in zebrafish, have rather described apical extrusion of RasV12 cells (see for instance work from Y Fujita lab). While I do understand that basal extrusion may have been overlooked in these other studies, could the authors speculate on what could explain the dominant role of basal extrusion in their system (especially compared to Takeushi et al. Curr Biol 2020 where the genetic system seems to be exactly the same) ? It would be fair to have some quick mention of this in the discussion.

- This is a point that is important but we do not feel we have space to go into this in full detail here. Instead, Yasu Fujita and I are planning to write a review or snapshot soon that will discuss all the different types of extrusion and the signaling that drives each. Here, we can simply state that KRas mutations, which we use, behave very differently to HRas mutations, which Fujita uses. HRas cells tend to get eliminated apically through a process that looks similar to apical extrusion but requires a slightly different mechanism and signaling process requiring prior cell division. We believe that this may well contribute to why cancers driven by HRas mutations are generally treatable and those (like pancreatic, lung, and colon) driven by KRas mutations are generally far more aggressive. We would prefer to give this aspect a much longer and scholarly treatment than is available in this format.

2. Many data/movies are rather hard to interpret and it is sometimes hard to connect the interpretation/description in the manuscript with the pictures/movies. Could the author first show systematically separated channels both in figures and movies ? (especially for Fig 4A to G and movie S7, movie S13). For instance, this is very difficult (if not impossible) to see the apical constriction on the lateral view provided in Fig. 4F (where is the E-cad belt of the neighbouring cells ? Where is the constricted apical side ?)

- We are puzzled by this request. In Fig. 4A-D, there is no overlap between E-cadherin with EGFP-KRas, as all the cells have been internalized. The case of GFP-CAAX, both

are co-expressed in the same cells where it hasn't invaded, for contrast. We think splitting the channels here would take up too much space. However, we have separated the magenta and green channels of E-cadherin and GFP-KRas as well as showing the overlay in Fig. 4G, which we think is essential for showing that E-cadherin lies above and gets pinched off. Of all the XZ sections, we think that this is the most telling, as it is one of the few times we were able to catch a cell in the act of BCE (which happens rapidly and stochastically), where we stained for E-cadherin. Here you can see how E-cadherin is separated at the time of BCE. In the stills of still of Movie 15 (Fig. 4F) when we separated the channels, you lose all useful information, as you really need the juxtaposition of actin below to indicate that actin is constricting below the KRas.

- In Fig. 4F, this is a single slice through the junction where the cell is invading, not a cross section of all epithelia around it. This point may become more clear, when you inspect what a slice through the monolayer looks like at different points, which we have included in Fig. S5F. At the point where we show E-cadherin colocalizes and is also slightly above the actin ring as specific points, there is no E-cadherin nearby to compare this to. We hope that this will make more sense.

3. The connections between basal extrusion, cell migration and tissue invasion remain quite indirect at this stage. So far, this is mostly based on one movie which show one event of extrusion and some basal body moving around and the link between perturbations reducing basal extrusion and the proportion of ectopic cells in the embryo.

- We have added more movies (S16 & 17), showing more examples of what we see in all cases: we see EGFP-KRas^{V12} in a NTR-mCherry (cytoplasmically localized) background pinches off green as cells basally extrude, enabling the red cells to migrate directly from the extrusion sites. Aside from these four movies (with Smovies 8&15), we also state that all cells surviving after BCE migrate away from the original site. At the beginning of page 4 we state, "***This severing of the apical membrane explained why all cells invading by BCE experienced concomitant membrane blebbing and loss of green fluorescence before migrating from sites of extrusion (e.g. Movies S8, 16, & 17 Fig. 3A'). Moreover, E-cadherin colocalizes with and slightly above apically localized actin in the periderm and becomes constricted into an apical point above a cell during BCE (Fig. 4G and SFig. 5F), suggesting it is also stripped from the invading cell***³³.

First, could the authors exclude that the basal body shown in Fig. 3A and movie S7 is not an apoptotic fragment transported in some macrophage? In fact, there are quite many magenta positive particles moving around on the basal side. If possible, it might be useful to check that there is no nuclear fragmentation (as one would expect following apoptosis) and that those basal moving bodies do contain the Ras cell nucleus.

As mentioned in our response to Reviewer 1:

- We now include one (of several) movies (Movie S9) of KRas^{V12} cells in an mpeg:mCherry zebrafish line that labels macrophages and included confocal projections of zebrafish immunostained for macrophages with L-plastin (S. Fig. 2C), indicating that

there is overlap with either. The finding that macrophages do not engulf live invading cells and we do not see them co-staining for macrophages by immunofluorescence using an antibody to EGFP, which is pH insensitive, shows that the internalized cells we see migrating, dividing, and morphing are not within macrophages.

- Additionally, we also include Fig. S2A&B showing stills from Movie S7 and immunostaining for active caspase-3 indicating the phenotypes seen when a cell invaded by BCE dies. In the stills and the movie expressing a H2B:RFP, you can see that one of the dying cell remnants gets engulfed, causing rapid disappearance of the EGFP signal, due to quenching by low pH. (This transgene does not allow one to easily follow BCE, however.) This demonstrates that cells invading by BCE no longer fluoresce green following engulfment by macrophages. This engulfment and loss of fluorescence is a standard response to the minority of invaded cells that die is easily distinguished from cells that continue to fluoresce EGFP following invasion by BCE.
- Finally, inclusion of two more movies (16 & 17) show that cells pinch off EGFP-dt-KRasV12 but continue to express cytoplasmic NTR-mCherry, as they migrate away. If these migrating cells were a product of macrophage engulfment, one would see red cells migrate to them, and engulf them prior to migrating, but this does not happen.

Similarly, in Fig. 4F, it is not clear how the authors can connect without ambiguity the Ras extruding cell with the magenta positive particle on the basal side (shown on the last panel, bottom right). Documented tracking in 4D would help here.

- This is a zoomed in movie of this process that we would prefer to keep, as it nicely shows the pinching off of the apical membrane. The remaining red cell goes out of focus and then back into focus, however, so is not as great at showing this point. Instead, we have included two movies showing invasion at lower magnification where it is easy to follow cells invading and migrating away from extrusion sites as they lose green and become red in Movies S16 & S17.

Finally, it would be really useful if the authors have captured unambiguously events connecting basal cell extrusion, cell migration, cell mitosis, access to the blood flow and/or abnormal differentiation in single movies. I do appreciate that it might be rare events difficult to catch and track on single embryos, but without that, it will be difficult to clearly connect these different cell populations unambiguously.

- The movies provided in S16 & 17 show cells migrating away from extrusion sites. While two authors had access to excellent light sheet microscopes that could follow long intervals, they could only access this microscope occasionally, so the bulk of our movies could not follow cells for the times required to detect all of these events. However, our next paper will address in more depth, using longer movies, how invading cells morph over time. We are finding some very exciting factors that will require another entire paper's worth of data to show convincingly.

4. The severing of the apical domain and the basal extrusion are quite interesting. On the first sight, the multiple blebbing and cell bodies may look like apoptotic fragments. As indicated

above, It would be really useful to provide movies of these basal extrusion with a marker of nuclei to show that the main cell body is basal and to exclude any nuclear fragmentation. This would also help to define the localisation of the cell among all these remaining debris (see above)

- While we have made many movies with a red marker of nuclei, these movies do not contain actin, however, needed to convincingly show that these cells invade by BCE. Nonetheless, we believe that the data that we provide showing a cell dying following basal cell extrusion and the lack of colocalization with any macrophages should rule out this possibility.

5. The title of some graphs and the associated legends are quite often ambiguous and it is not always clear what is exactly represented (especially for graph showing percent of something). This includes:

Fig 3G : is the quantification in p53 background ?

- Yes, only Fig. 1 is in a wild type background. Thanks for pointing this out. All figure legend titles reflect this now.

Fig 3I: What represents the percent of cells is not so clear, is it the percent of all GFP positive cells expressing N-cad or only the cell internalized (with one dot = one embryo) ? Is this percentage different for the internalized Ras cells and the Ras cells in the epidermis ?

- Yes, we now say, '**Percentage of invaded *dt-KRas*^{V12} cells that are N-cadherin-positive per embryo, as mean \pm SEM.**'

Figure 4E: Is it the percentage of embryos showing snail positive cells in the different localisation or is that the proportion of Snail positive cells for the full Ras population of a given compartment in one embryo ?

- The latter, the proportion of Snail-positive cells within KRas embryos in each location.

6. Fig 4HI : is the total number of surviving Ras V12 cell similar in the controls in the ROCKOUT ? There seems to be much less cells in ROCKOUT treatment (while one would expect to find more cell in the epidermis in this condition). One alternative explanation for the absence of internalized cell could be that this factor affects cell survival after extrusion (and not the exit from the layer).

- There are more cells in the epidermis. This is not what we quantified here, though. Only internalized cells to internalized cells (y-axis). You can see in SFig. 5H that there are more KRas cells in the epidermis.

Other minor points:

- The font size seems to change in the middle of page 3
Changed.

- Fig 2D: are these global morphological defects commonly observed upon depletion of p53 and induction of RasV12 in a subset of cells ? Do the deformations always correlate with the presence of Rasv12 cells masses ?

- Yes, we do frequently note abnormal looking embryos when they have large internalized masses. They do not look well.

- Fig. 2B: How do the authors identify Ras dying cells unambiguously ? I see that the authors mention cleaved caspase3 staining and morphological features in the methods. It would be useful to give more details and maybe show a representative example of each category.

- Yes, we have added another Supplementary Figure (2B) to address this and added this to the text: “Although EGFP-KRas^{V12} BCE caused cells to internalize, many died (Fig. 2B), **as detected by their fragmentation and rapid engulfment in movies (SFig. 2A & Movie S7) or by active caspase-3 immunostaining (SFig. 2B).**”

•

- In the legend of Fig. 3E, the number of embryos is missing

- I think you mean 3I and we have added this now.

- There is a rather large literature in Drosophila documenting how the combination of RasV12 activation with various mutations lead to invasive behaviours (check for instance numerous works from Tian Xu lab, Yale School of Medicine). Although to my knowledge this was never associated with long term live imaging, it might be relevant to compare the behaviour observed in zebrafish with these works (including similarities and differences).

- This is an interesting point but different, as cells in Drosophila typically extrude basally by default. The Xu paper investigates mutations that instead cause cells to extrude apically. This may be an interesting topic for another review but we feel that it may be too abstract to discuss in the short format of the paper here.

Reviewer #3 (Remarks to the Author):

KRas-transformed epithelia cells invade and partially dedifferentiate by basal cell extrusion.
Fadul et al.

In this paper, Fadul and colleagues describe the phenomenon of basal cell extrusion (BCE) as a mechanism by which Kras-transformed epithelial cells leave an epithelium and disseminate to distant sites. This work stems from previous work in the lab, which has defined cell extrusion as a process to remove excess or dying cells from epithelial tissues without breaching barrier function. Here, they build on previous work and combine zebrafish in vivo models with live cell imaging to study cell extrusion in real time. They find that as well as triggering the formation of neoplasia, Kras-transformation induces BCE of cells, independent of neoplasia. Interestingly, BCE is associated with a change in cell morphology of the extruded cells, leading to the speculation that dissemination is linked to a change in cell fate. Moreover, the authors show that extruded cells are unable to survive unless they lose functional p53. Kras mutations are key driver mutations in pancreatic and lung cancers, both of which are highly metastatic and of unmet clinical need. In addition, there is increasing evidence that metastasis occurs at very early time points and before overt tumour formation, challenging our understanding of metastasis as a progressive step in tumorigenesis. The findings in this paper are novel and

important as they support this current view and shed light on the mechanisms underlying early metastasis. While this paper is very interesting and compelling, it requires revision before being considered for publication at Nature Communications. I hope my comments will help to shape discussion and interpretation and improve clarity, particularly for the non-expert.

- Thanks for your comments and providing these helpful points.

The main weakness of the paper is that it is not clear whether there is a distinct molecular mechanism(s) underlying how KrasV12 cells decide whether to be extruded apically or basally or overgrow in cell masses, or whether each phenomenon is stochastic. The authors demonstrate a number of mechanisms are required; however, as written it is not clear how each mechanism works together in the system. Is BCE stochastic or specific to a region of the tissue?

- The regions in Fig.1 C&F show that where cells invade by BCE are central, where cells typically divide and that where they form masses are at the fin edges, typically where cells extrude, as defined in our 2012 Nature paper. Interestingly, a recent paper we published in collaboration with the Horvitz lab shows that cells in replicative stress are more prone to extrude, offering a potential mechanism for why some cells are more likely to basally extrude. We cite this now in the middle of page 2, "**Extrusion in division zones may be due to high replicative stress and DNA damage**"²².

How does lack of functional p53 contribute to the change in Kras cell fate/morphology or is it only required to increase survival?

- Our findings suggest that it does not dramatically affect extrusion but only survival, as many of the same internalized cell types occurred in a WT background but did not survive without p53 loss. We state at the bottom of page 4, "**We find that KRas-transformation directly causes cell invasion, while p53 mutation enables their survival.**"

What is the requirement of cell proliferation/division in the system?

- We frequently see proliferation of internalized cells and expect it to contribute to the large internalized masses over time. It would be hard to test this directly, as blocking cell division for 3 days would kill the developing embryo.

Some clarity/discussion on each of these points would strengthen the message.

Minor points to address:

1. Fig 1D illustrates apical and basal extrusion – judging from the IF images, the process is similar in both and is only distinct by the direction of extrusion. Have the authors confirmed direction of extrusion using defined apical and basal markers? Here the authors are tracing GFP yet describe contracting F-actin rings. Do they have data staining for F-actin or similar to support this conclusion? Later they show that inhibition of autophagy inhibits BCE (using chloroquine) Under these conditions, are KrasV12 cells apically extruded instead? This would also help understand mechanism.

- Live actin markers showing the ring contracting above in a basal extrusion are in Figs. 3A and 4F. We have also investigated apical and basal markers in our first study on this, using cell culture, where we also discovered that blocking autophagy flips the direction of extrusion. We found it does so by blocking degradation of the rate-limiting lipid ligand, S1P. Once S1P is rescued, it rescues apical extrusion (Slattum et al, Current Biology, 2014).

2. Experiments are based on mosaic expression of eGFP-KrasV12; however, it is not clear what mosaic expression means in terms of neighbourhood. Can they estimate how much of the tissue is expressing the transgene? Does BCE (and apical extrusion) occur from regions of the epithelium that is predominately wild-type (i.e., where mutant cells have normal neighbours) or are labelled cells being extruded from clones/clusters of transformed cells?

- In our previous Current Biology paper, we show that KRas^{V12} expression acts cell autonomously to drive basal extrusion. Additionally, there is growing evidence from Alpha Yap's, Sarah Woolner's lab and others, to suggest that differential contraction rates, intrinsic to differential expression of KRas here, promotes extrusions. Therefore, while we have not studied this here, we suspect that KRas^{V12}-expressing cells will be far more likely to extrude when surrounded by wild type neighbors than mutant KRas neighbors.

3. Fig. 1B describes cell masses per embryo and relates to main text "masses of 3-40 KrasV12 cells formed..." what is the timeline here? What time post fertilisation does Fig. 1B relate to? A definition of 'hpf' in the figure legends would help the non-expert. The maps in Fig. 1C and 1F suggest that cell masses form in early embryos; whereas extrusion occurs later in development. Please comment.

- These are at 26 hpf, as mentioned in the figure legend and we have now ensured that all figure legends state the age of embryos used. The masses do form early and it seems like many resolve as the embryo grows and stretches by 48-56 hpf. The extrusions occur throughout development but, as you can see from our diagrams in Fig. 1F, appear more frequent later in development. I imagine that there would be even more later in development, if we had inducibly expressed KRas later. However, in our reported experiments most epidermal cells have already extruded by 4-5 dpf.

4. The authors describe how cells expressing c-Myc did not form masses, are not extruded or internalised (top of Page 4). This is assuming that KrasV12 cells are internalised; however, this appears very suddenly, and it is not clear which of the data describes internalised Kras cells. My interpretation is that KrasV12 cells invade the basal epidermal layer; yet as written the authors suggest that transformed cells are encapsulated in other cells. Please clarify with supporting evidence. If it is indeed that extruded cells invade the basal layer, I would argue that 'internalised' is misleading here and needs to be reconsidered as a description in subsequent text/figures.

- Thanks. We have changed this to invaded. Here, we instead state: "**By contrast, cMyc over-expression, implicated in metastasis²⁴, did not cause masses, extrusion, or**

invasion, suggesting that not all oncogenic signalling drives BCE or invasion (Fig. S1C,D & Movie S6)."

5. The authors conclude that overexpression of oncogenic c-Myc upregulates growth and proliferation and this has no effect on BCE, transformed cell mass or invasion. Do the authors have evidence to show with confidence that c-Myc overexpression does indeed increase growth/proliferation in vivo? Overexpression of PI3K signalling may be a better control here. This is important because the data demonstrate that extrusion occurs at sites of cell division; therefore, increasing cell division rates may increase extrusion and provide a possible molecular mechanism for extrusion.

- Thanks for pointing this out. This was not really a point we wanted to emphasize and instead have changed the text to the above to state that not all oncogenic mutations lead to BCE and invasion, as quoted above. This is also true of p53 mutation.

6. The authors claim that transformed cells that are basally extruded and survive, "adopt new plasticity". The authors use acetylated tubulin and N-cadherin to define the new cell states, concluding that the cells adopt neuronal/mesenchymal fate. My interpretation is that the authors chose these markers based on changes in cell morphology rather than plasticity. The term 'plasticity' suggests a reprogramming to a stem-like fate. Have the authors screened for additional markers in these cells to conclude that the cells adopt a plastic state; e.g., markers of stemness or EMT markers. E.g., Zeb-1 has been shown to be a marker of early circulating tumour cells in pancreas. Indeed, they show that cells enter the circulation; are circulating cells also stem-like?

- These are good points. We did try to stain for these markers by immunofluorescence and in situ hybridization with no luck. We think a better approach to this may be to do transcriptomics on all internalized cells followed by principal component analysis. This will require us to make a stable CRE line to do this properly and will be the basis of another paper in which we identify the different cell types derived from BCE and factors that contribute to their genesis. We do not have the space to do this properly in this first paper.

7. Top of page 7 – the authors conclude that BCE cells lose epithelial markers but retain mesenchymal markers. This is based on the fact that expression of eGFP is from the keratin 4 promoter (not clear why they refer to Fig. 2E to support this). In my view, an additional (EMT) marker is required to show mesenchymal cell status and support this conclusion.

- Here we are only referring to the fact that cancer cells can express both epithelial and non-epithelial markers. We hope that the revision makes more sense: "***While much of EGFP signal within invading KRas^{V12} cells is initially pinched off, its re-expression from the keratin 4 promoter, suggests that cells invading by BCE continue to express this cytoplasmic epithelial marker (Fig. 2E). Thus, cells invading by BCE lose cell surface epithelial determinants but retain cytoplasmic ones, with only some adopting mesenchymal features. These findings could address why metastatic tumours can express both epithelial and mesenchymal markers^{30,31}.***" In

our case, only some cells clearly express mesenchymal markers, N-cadherin and Snail1b.

8. Mechanistically, the authors show that loss of E-cadherin is not likely to be dependent on changes in transcription or dynamin-mediated endocytosis. They present compelling data to show that E-cadherin is lost via pinching of an actin ring during extrusion process. Inhibition of Rho kinase (ROCK) also blocks BCE and invasion. While the authors claim that this is distinct from extrusion of v-Src-expressing cells (Anton et al., 2018), do the authors have sufficient evidence to support this claim; is myosin activity also required in the actin ring formation and pinching?

- We used these inhibitors specifically because we know that actin and myosin are required for extrusion (Rosenblatt Curr. Biol. 2001, Slattum, JCB 2009) and have cited two papers that show it blocks actomyosin contraction in zebrafish. Again, it blocks invasion by BCE. We are not sure what alternative hypothesis would account for this.

9. The schematic in Fig S5 is misleading as cancer cell invasion occurs from a cluster of transformed cells that are also apically extruding. In the text (final paragraph page 10) the authors claim that BCE occurs independent of cell masses, which is not illustrated in the cartoon. This comes back to my first point (where in tissues are KrasV12 cells being basally eliminated from) and needs clarification.

- Here, we show on the bottom transformed section on the left there is a mass (perhaps this is what you consider apical extrusion?) and on the right side of the epithelium, a single cell basally extruding. The basally extruding cell is shedding its apical domain (light yellow/green) with its cadherins, allowing the escape of the remaining portion of the cell beneath the epithelium, where it then can migrate, differentiate, and enter the bloodstream. This point that cells invade at sites distant to where masses form is the point behind our findings in Fig. 1C & F. We have now changed the text associated with this figure and hope that this now makes sense: **“Supplementary Figure 6. Model. Top panel: In wild-type epithelium, cells get extruded and die when they become too crowded. Bottom panel: Epithelia with an oncogenic mutation in KRas form masses at crowded sites (left) or invade under the epithelium by basal cell extrusion (BCE) at completely separate sites (right). BCE allows cells to invade and simultaneously pinch off their apical epithelial determinants, enabling migration, new plasticity, proliferation, and entry into bloodstream. While KRas-transformation enables invasion, most cells will die unless p53 is also mutated.”**

If not, please let us know how to change it.

REVIEWER COMMENTS>

Reviewer #1 (Remarks to the Author):

During the revisions, the authors have addressed many of the points raised in our first review, including the addition of new data. After these additions, the manuscript is considerably improved. However, some of the new data are presented in a rather confusing way, requiring (at least) editorial adjustments, while some other crucial additions are still missing.

1. Some clarifications are needed to fully support the claim that the GFP-positive EGFP-KRASV12 cells found inside the fish derive from peridermal cells.

1a. It is still unclear in the main text, methods, and in the figure legends how “misexpression” (which means experimental “noise” of plasmid injection) is distinguished from “internalised” (which means “real” periderm-derived cells after BCE). In line 56 of the main text, misexpression of GFP is described in the notochord, muscle, or melanocytes. Are these the only “misexpressing” cell types? And what is the rationale for this classification? Because similar numbers are found upon *krt4:EGFP-KRASV12* and *krt4:GFP-CAAX* control plasmid injection? In turn, are numbers of “internalised” cells calculated as the number of GFP+ cells below the periderm, minus the number of GFP+ notochord, muscle, and melanocyte cells? Please amend the text of the methods to clearly describe how “misexpressing” and “internalised” cells are identified, e and calculated.

1b. The authors have provided encouraging new data showing injection of UAS:EGFP-KRASV12 into wild-type versus ET(periderm:Gal4) transgenic fish (Figure S1B), to support the model of EGFP-KRASV12 cell invasion. However, the data are presented in a manner which is seemingly contradictory and we suggest to re-structure the figure. Figure legend S1 states that the authors found 30 internalised cells per embryo when UAS:EGFP-KRASV12 was injected into the ET(periderm:Gal4) fish, compared to only one internalised cell in 27 embryos when it was injected into negative control wild-type fish, nicely in line with the hypothesis that KRASV12-expressing cells in the periderm basally invade. On the other hand, the presented graph of the same experiment demonstrates that 55% of negative control WT injected embryos missing the Gal4 driver cryptically express the UAS:EGFP-KRASV12 construct. This is confusing and seemingly contradictory. Again here, a definition of internalised vs misexpressed would clear up a lot of confusion, combined with re-structuring of the graph, showing the numbers of GFP+ cells per fish and per category (periderm, misexpressing, or internalised).

1c. In Figure S4A, the neurons stemming from peridermal *krt4:EGFP-KRASV12* cells are GFP+, yet mCherry-. The authors have explained that the enhancer trap promoter (periderm:Gal4) driving UAS:lifeact-mCherry might be turned off following invasion. This seems like a reasonable argument, but the new experiment discussed above (Figure S1B), uses the same enhancer trap promoter, yet UAS:EGFP-KRASV12 is apparently still expressed following invasion. The authors have also now included two new videos (16 and 17) showing loss of GFP, but retention of UAS:NTR-mCherry using the same enhancer trap promoter, following invasion. These contradictory results weaken the argument that the invaded cells derive from the periderm and need to be addressed in the text.

2. As pointed out in our initial comments, we find it difficult to clearly see whether the movements of the invaded cell actually belong to a live cell or only cell remnants, possibly engulfed by an immune cell immediately following BCE. The cells undergoing basal cell extrusion pinch off a lot of vesicles, strongly resembling apoptosis. Therefore, it is really crucial to show by live imaging that a basally extruding cell first pinches off most of its content but is still alive after that. To show this, we had suggested to use a transgene to label nuclei (as the *h2afva:h2afva-mCherry* reporter line) and to actually follow the (intact) nucleus of the (living) extruded cell. In their revised version, the authors have now included a supplementary movie (7) and stills in supplementary figure 2 showing an event in which a basally extruding cell actually undergoes cell death and the cytoplasmic remnants are engulfed by another cell, which could possibly be a macrophage or also neutrophil. The authors mention that this happens in the minority of invaded cells, whereas the majority survives. However, they fail to show a nucleus of a surviving invading cell as “this transgene does not allow one to easily follow BCE”. To us, it is not clear why it should be impossible to follow a cell expressing cytoplasmic GFP from *krt4:EGFP-T2A-KRas* in combination with the nuclear marker. Even if the GFP signal fades away, it should be possible to track the nucleus of the initially green cell via time-lapse imaging when the time intervals are set small enough. Including such a movie of a *krt4:EGFP-T2A-KRas* cell with an intact nucleus moving away after BCE would greatly strengthen the authors’ conclusions about the invasive character of such *krt4:EGFP-T2A-KRas* cells. Therefore, we still think that this experiment is absolutely crucial. Why do the authors for instance not simply add movie 6 of their Biorxiv preprint (doi: <https://doi.org/10.1101/463646>)?

3. The hypothesis regarding mechanical stripping of apical determinants (like ZO-1 of tight junctions and E-cadherin of adherence junctions) during basal cell extrusion, although very interesting and one of the major new findings of the manuscript, still has not been fully supported. While it may be true that E-cadherin localizes with, and apical to, the constricting actin ring, it is additionally found along the entire lateral and basal compartments of peridermal cells of the embryonic zebrafish epidermis. The latter has been convincingly demonstrated in several previous publications, employing higher resolution images of cryosections through the epidermis than supplied in the current manuscript (for instance Arora et al 2020, *eLife*, Figure 1 B1-B5, and references therein).

With this in mind, the authors must explain the complete loss of E-cadherin from basally extruding cells, given that only a small portion should be lost when the apex of the cell is pinched off. The authors themselves state in the rebuttal letter, “If it were basolateral, as suggested, it would not disappear.” That is precisely the problem. It IS basolateral in peridermal cells, so why does it disappear? The authors need to revise their model accordingly, for instance integrating additional mechanisms such as a preceding re-distribution of E-cadherin from basolateral to apical domains of *EGFP-KRASV12* cells or an concomitant loss of E-cadherin by other mechanisms that occurs in parallel to apical stripping.

4. Figure 3I: we still don’t understand why the GFP-CAAX numbers are not included. There are also “hardly any” neuron-like cells in the GFP-CAAX category in Figure 3G, yet this WAS included. Also, the figure legend still does not explain this. Please include the data of the GFP-CAAX injected fish in Figure 3I, to match the GFP-CAAX data included in Figure 3G, and amend the text of the figure legend.

5. The *snai1b* labelling experiment is still a bit unclear and should be clarified. We still have some of the same questions, which were not answered in the authors’ rebuttal.

5a. Regarding the ectopic *snai1b* expression, in the methods, ectopic expression is defined as compared to uninjected embryos, whereas in the rebuttal letter, it is now referred to expression of

snai1b in CAAX injected embryos. Either way, where is normal, endogenous snai1b expression observed? Please correct the methods to reflect that this was performed in CAAX injected embryos. If the GFP-CAAX fish are the guide for normal, endogenous expression, how do you define “ectopic”, and why are there 20% of the GFP-CAAX fish with “ectopic” cells (Figure S4D)? This is still confusing and makes it very difficult to understand the results in the EGFP-KRASV12 fish.

5b. What has also not been answered from our initial questions, is whether all of the “ectopic” snai1b+ cells are double positive for snai1b and EGFP, and what is the percentage of double-positive cells compared to total invaded EGFP-positive cells? In Figure S5E, example images are shown of snai1b+/EGFP+ or EGFP+ only cells, but this is not quantified anywhere. In line 89 of the main text, it is stated “ Additionally, ~15% of invaded cells express the mesenchymal marker, N-cadherin (Fig. 3H&I) and mesenchymal transcription factor, snai1b (Fig. S4C&D).” It is not clear if cells with ectopic snai1b expression belong to the same population as the N-cadherin positive cells from this sentence. Are 15% of invaded cells N-cadherin positive, another 15% snai1b positive? The quantification for snai1b is missing here, please amend the text.

5c. Minor comment: Anterior and posterior in Fig S4C might actually be dorsal and ventral. If so, this should be corrected.

Reviewer #2 (Remarks to the Author):

The authors have addressed part of my concerns by providing new movies and further explanations. Most of the argument provided by authors in the text and the rebuttal to exclude alternative hypothesis (leaky labelling, transport by macrophages) are overall very reasonable but I feel that most of them are still debatable (see below). I believe that the manuscript still lack some critical experiments that would really remove any ambiguity regarding the fate of the active Ras cells, including movies with nuclei labelling and long term fate tracking of cells. Because the fate of the Ras cells is the essential message of this article, I think it would really deserve proper fate mapping/tracking.

Since the latter is not possible through long term live imaging, and that the Cre system is not available now, I do believe there might be alternative based for instance on photoconversion of some epidermal Ras cells and localization/morphological description of the photoconverted cells one or two days later. This would solve the questions related to potential leakage of the construct and the origin of the internal cells. I do appreciate that this experiment will not be straightforward, but this would solve once for all the question of the fate of the Ras cells.

Regarding the arguments provided by the authors regarding macrophage contribution in the rebuttal, while they are all reasonable, I am not sure they totally exclude that Ras cell bodies movements is driven by them. For instance, in movie S9, while the GFP is clearly away from macrophage at the beginning, there is an overlap with purple signal (albeit fainter) when the GFP signal starts to move

around. Similarly, the argument about the red labelling of macrophage (if they have already ingested before other labelled cells) is correct provided that the majority of macrophage have already engulfed labelled cells (which depends on the number of macrophage and the proportion of labelled cells). If there is a significant proportion of macrophage that have not yet engulfed labelled cells, the argument would not hold.

Regarding ectopic labelling, while the number of embryos with ectopic labelled cells is the same in the CAAX and the Ras embryo (Figure S1B), nothing exclude that the transfection of Ras + p53 deletion favor the survival and proliferation of the ectopic labelled cells (as pointed out by referee 1), which could explain the higher number of internal cells in this context. So while the interpretation proposed is plausible and reasonable, I don't think it is possible now to fully exclude the contribution of ectopic labelling unless there is a proper tracking and/or fate mapping.

Other minor points:

- Figure S1B: Could the authors provide details on how the transfected cells were tracked in absence of Gal4 ? I guess there was another fluorescent label transfected but this could help to specify in the legend. Also, could they provide more details on what they defined as misspecified (I suspect non-epidermal but this should be stated somewhere) ? Also I would have expected this number (proportion of embryo with ectopic cells) to be higher in RasV12 cells compared to CAAX cells (right part of the graph) specially if Ras promote cell internalisation. Actually this quantification could be compatible as well with Ras transfection promoting survival and proliferation of ectopic labelled cells (same initial number of ectopic labelled cells with the control, but higher cell number at the end because of higher proliferation and survival).

- It was not clear to me where were the invading cells in movie S4 among all the GFP basal round bodies. Could the authors point to the relevant structure in the movie ?

- Figure S2C right: could the authors indicate on the panel where are the extruded Ras cells ?

- When the authors refer to the neuron like morphology, they do observe rare case of neuron like morphology in the CAAX control. Would that be a typical exemple of ectopic labelling ?

- Figure 3H: what are these other labeled N-cad structures which are not GFP positive ?

- Figure 4F: Since there is no clear evidence that the purple label structure at the bottom comes from the extruding cell (purple arrow) I would not indicate it on the figure and the legend unless the authors can unambiguously track it (which I guess they cannot if it is out of the frame for several time frame).

Reviewer #3 (Remarks to the Author):

This study is novel and interesting to a broad readership. It is of a sufficient standard to warrant publication at Nature Communications.

Reviewer #1 (Remarks to the Author):

During the revisions, the authors have addressed many of the points raised in our first review, including the addition of new data. After these additions, the manuscript is considerably improved. However, some of the new data are presented in a rather confusing way, requiring (at least) editorial adjustments, while some other crucial additions are still missing.

1. Some clarifications are needed to fully support the claim that the GFP-positive EGFP-KRASV12 cells found inside the fish derive from peridermal cells.

1a. It is still unclear in the main text, methods, and in the figure legends how “misexpression” (which means experimental “noise” of plasmid injection) is distinguished from “internalised” (which means “real” periderm-derived cells after BCE). In line 56 of the main text, misexpression of GFP is described in the notochord, muscle, or melanocytes. Are these the only “misexpressing” cell types? And what is the rationale for this classification? Because similar numbers are found upon krt4:EGFP-KRASV12 and krt4:GFP-CAAX control plasmid injection? In turn, are numbers of “internalised” cells calculated as the number of GFP+ cells below the periderm, minus the number of GFP+ notochord, muscle, and melanocyte cells? Please amend the text of the methods to clearly describe how “misexpressing” and “internalised” cells are identified, e and calculated.

We have now clarified this in the methods (**blue**, lines 285-7 & 292-4). Classic misexpression in transient F0 transgenics occurs mostly in muscle, but also in notochord and melanocytes in both KRas^{V12}- and CAAX-injected embryos (see the revised SFig. 1 and our response to 1b, immediately below). Yes, the number of invading cells are calculated as the number of internalized cells (no longer in the periderm) minus the misexpressing cells. Therefore, misexpressing cells are excluded from our graphs and statistical analyses, as stated in the Methods (**blue**, lines 285-7).

1b. The authors have provided encouraging new data showing injection of UAS:EGFP-KRASV12 into wild-type versus ET(periderm:Gal4) transgenic fish (Figure S1B), to support the model of EGFP-

KRASV12 cell invasion. However, the data are presented in a manner which is seemingly contradictory and we suggest to re-structure the figure. Figure legend S1 states that the authors found 30 internalised cells per embryo when UAS:EGFP-KRASV12 was injected into the ET(periderm:Gal4) fish, compared to only one internalised cell in 27 embryos when it was injected into negative control wild-type fish, nicely in line with the hypothesis that KRASV12-expressing cells in the periderm basally invade. On the other hand, the presented graph of the same experiment demonstrates that 55% of negative control WT injected embryos missing the Gal4 driver cryptically express the UAS:EGFP-KRASV12 construct. This is confusing and seemingly contradictory. Again here, a definition of internalised vs misexpressed

would clear up a lot of confusion, combined with re-structuring of the graph, showing the numbers of GFP+ cells per fish and per category (periderm, misexpressing, or internalised).

We recognize now that the previously requested data on misexpression versus invading cells only served to confound our findings, making it seem that most invaded cells stem from misexpression. They do not. To clarify this, we have now quantified cells per fish into misexpression categories (muscle, notochord, or melanocyte,) **compared to** invaded cells in SFig. 1B. The new graph clearly shows that there are far more invaded cells than mis-expressing cells in UAS:EGFP-KRas^{V12}-injected into periderm-Gal4 but not in WT embryos. While there are more mis-expressing cells in the wild-type, we only found one potential invaded cell, indicating that invading cells **are not linked** to misexpression. To make it as clear as possible what we categorize as mis-expressing cells, we also provide examples of misexpression in the muscle, notochord, or melanocytes in SFig. 1C, as well as zoomed out examples of whole zebrafish tails from control wild-type or periderm-Gal4 fish (SFig. 1D). We scored invaded cells as those not at the epidermis or in these mis-expression categories. We have updated the figure legend title for this in **blue** as well. While the Gal4 driver typically expresses fewer GFP-positive cells than does the KRT4 driver, used in the majority of the paper, it produces a similar high ratio of invaded to misexpressed cells. The similar number of mis-expressed cells stemming from KRT4:KRas^{V12} are far less than those expressed within the periderm (see Fig. 2C), further reducing the likelihood that any cells scored as 'invading' stem from mis-expression. We believe that this quite clearly shows that muscle or other cells that misexpress KRas^{V12} do not contribute to the invaded cells we scored.

1c. In Figure S4A, the neurons stemming from peridermal krt4:EGFP-KRASV12 cells are GFP+, yet mCherry-. The authors have explained that the enhancer trap promoter (periderm:Gal4) driving UAS:lifact-mCherry might be turned off following invasion. This seems like a reasonable argument, but the new experiment discussed above (Figure S1B), uses the same enhancer trap promoter, yet UAS:EGFP-KRASV12 is apparently still expressed following invasion. The authors have also now

included two new videos (16 and 17) showing loss of GFP, but retention of UAS:NTR-mCherry using the same enhancer trap promoter, following invasion. These contradictory results weaken the argument that the invaded cells derive from the periderm and need to be addressed in the text.

We are not sure why the actin expression is lower here, however, it still is expressed. We recognize that making our figures accessible to colour blind readers, of whom we have a very vocal advocate in our lab reduces the ability to detect overlap in magenta/green. Embedded below is the original picture of this in red and green, where you can see that this neuron is yellow, expressing both lifeact-mCherry and GFP-KRasV12. The red actin may not be as bright in the fine neurite-like processes, similar to what we find of googled images of actin in in vivo neurites. Additionally, as stated before, the different

promoters may cause unequal expression of KRas v. actin.

2. As pointed out in our initial comments, we find it difficult to clearly see whether the movements of the invaded cell actually belong to a live cell or only cell remnants, possibly engulfed by an immune cell immediately following BCE. The cells undergoing basal cell extrusion pinch off a lot of vesicles, strongly resembling apoptosis. Therefore, it is really crucial to show by live imaging that a basally extruding cell first pinches off most of its content but is still alive after that. To show this, we had suggested to use a transgene to label nuclei (as the h2afva:h2afva-mCherry reporter line) and to actually follow the (intact) nucleus of the (living) extruded cell. In their revised version, the authors have now included a supplementary movie (7) and stills in supplementary figure 2 showing an event in which a basally extruding cell actually undergoes cell death and the cytoplasmic remnants are engulfed by another cell, which could possibly be a macrophage or also neutrophil.

The authors mention that this happens in the minority of invaded cells, whereas the majority survives. However, they fail to show a nucleus of a surviving invading cell as “this transgene does not allow one to easily follow BCE”. To us, it is not clear why it should be impossible to follow a cell expressing cytoplasmic GFP from krt4:EGFP-T2A-KRas in combination with the nuclear marker. Even if the GFP signal fades away, it should be possible to track the nucleus of the initially green cell via time-lapse

imaging when the time intervals are set small enough. Including such a movie of a krt4:EGFP-T2A-KRas cell with an intact nucleus moving away after BCE would greatly strengthen the authors' conclusions about the invasive character of such krt4:EGFP-T2A-KRas cells. Therefore, we still think that this experiment is absolutely crucial. Why do the authors for instance not simply add movie 6 of their Biorivx preprint (doi: <https://doi.org/10.1101/463646>)?

As we explained before, we *have imaged* invasion in a zebrafish line where nuclei fluoresce red (as included in our supplemental movie). These movies do not address whether cells invading by BCE die, because **they do not allow us follow actin ring contraction, which is a necessary hallmark of BCE**. Without it, we cannot determine whether cells have basally extruded or not, so this will not answer the question you ask. We have followed numerous movies and cannot tell ourselves. To make another zebrafish line that expresses both red actin and red nuclei would not only take over a year to produce, and would also confound analysis, as we would be unable to determine the difference between red actin blebs and red fragmented nuclei.

Instead, we have gone to considerable effort to address that migrating, invading cells are not a product of macrophage engulfment using five separate, and far more rigorous methods:

- (1) They do not colocalize with mpeg:mCherry-labeled macrophages in live imaging.
- (2) They do not colocalize with macrophage markers in fixed samples stained with a chicken anti-GFP antibody, where GFP **does not** disappear from the acidification associated with phagocytosis, as they would in live movies. Importantly, if the GFP did disappear, we would never have scored them in the first place, so this is already a moot point.
- (3) For contrast, we show an example (SFig. 2a and SMovie 7) of an invaded cell that dies and IS ingested where GFP disappears, as expected. This dying cell's fragmentation, ingestion, and loss of GFP is **completely different** to what we see in all our invasion movies.
- (4) We include several additional movies of green KRas^{V12} cells remaining for hours and migrating following extrusion—events that we could not have observed if macrophages engulf live invaded cells, like they do when cells die, since GFP disappears. You also need to consider: how might cells divide or differentiate, if dead (or even live) within a macrophage?
- (5) We show that few of the immunostained invaded cells are caspase-3-positive, also ruling out death and engulfment as a mode for their later migration.

3. The hypothesis regarding mechanical stripping of apical determinants (like ZO-1 of tight junctions and E-cadherin of adherence junctions) during basal cell extrusion, although very interesting and one of the major new findings of the manuscript, still has not been fully supported. While it may be true that E-

cadherin localizes with, and apical to, the constricting actin ring, it is additionally found along the entire lateral and basal compartments of peridermal cells of the embryonic zebrafish epidermis. The latter has been convincingly demonstrated in several previous publications, employing higher resolution images of cryosections through the epidermis than supplied in the current manuscript (for instance Arora et al 2020, eLife, Figure 1 B1-B5, and references therein).

With this in mind, the authors must explain the complete loss of E-cadherin from basally extruding cells, given that only a small portion should be lost when the apex of the cell is pinched off. The authors themselves state in the rebuttal letter, "If it were basolateral, as suggested, it would not disappear." That is precisely the problem. It IS basolateral in peridermal cells, so why does it disappear? The authors need to revise their model accordingly, for instance integrating additional mechanisms such as a preceding re-distribution of E-cadherin from basolateral to apical domains of EGFP-KRASV12 cells or an concomitant loss of E-cadherin by other mechanisms that occurs in parallel to apical stripping.

The contention that E-cadherin is basolateral is at odds with the entire epithelial field. Although you suggest others have shown it, you refer to one paper that *presumably* contradicts the field and importantly here, **our data**. Although not comprehensive, we provide a few reviews and primary papers that clearly demonstrate that E-cadherin localizes to the apical cell cortex in a variety of species (*C. elegans*, zebrafish, and mice):

reviews

- The assembly and maintenance of epithelial junctions in *C. elegans*
<https://www.fbscience.com/Landmark/articles/pdf/Landmark3316.pdf>
- Cell Polarity and Migration: Emerging Role for the Endosomal Sorting Machinery
<https://journals.physiology.org/doi/full/10.1152/physiol.00054.2010>

primary papers

- Control of E-cadherin apical localisation and morphogenesis by a SOAP-1/AP-1/clathrin pathway in *C. elegans* epidermal cells
<https://journals.biologists.com/dev/article/142/9/1684/47252/Control-of-E-cadherin-apical-localisation->
- Interplay of MPP5a with Rab11 synergistically builds epithelial apical polarity and zonula adherens
<https://journals.biologists.com/dev/article/147/22/dev184457/226095/Interplay-of-MPP5a-with-Rab11-synergistically>
- Sfrp Controls Apicobasal Polarity and Oriented Cell Division in Developing Gut Epithelium
<https://www.ncbi.nlm.nih.gov/pmc/articles/PMC2649445/>

A short consideration of the premise that E-cadherin is instead basolateral leads to the view that epithelia, instead of forming continuous flat sheets that protect tissue against the outer environment, form a sort of bubble wrap that would not be protective. This is simply not true, based on observation of many epithelial types, and defies the primary function of all epithelia. Most importantly, here, we **provide data** in **Supplementary Figure 5** that very clearly demonstrates that E-cadherin localizes apically in the outer epidermal layer with and above the apical cortical actin, in agreement with the rest of the field. The single paper cited (Arora et al., eLife 2020) does not show this. Instead, they show that E-cadherin is potentially basal to **OTHER** markers (not actin, the issue here). However, they do not provide XZ sections showing co-localization with these markers (which we provide), necessary to make this claim. While the underlying basal layer of cells (which do not form a tight barrier on their own) contain basolateral E-cadherin, this is not true of the periderm where we express KRas^{V12}. Therefore, we believe that the data we provide (and that from the rest of the field) clearly show that E-cadherin localizes apically at a location where it could be stripped off with basal cell extrusion.

4. Figure 3I: we still don't understand why the GFP-CAAX numbers are not included. There are also "hardly any" neuron-like cells in the GFP-CAAX category in Figure 3G, yet this WAS included. Also, the figure legend still does not explain this. Please include the data of the GFP-CAAX injected fish in Figure 3I, to match the GFP-CAAX data included in Figure 3G, and amend the text of the figure legend.

We have now added this.

5. The *snai1b* labelling experiment is still a bit unclear and should be clarified. We still have some of the same questions, which were not answered in the authors' rebuttal.

5a. Regarding the ectopic *snai1b* expression, in the methods, ectopic expression is defined as compared to uninjected embryos, whereas in the rebuttal letter, it is now referred to expression of *snai1b* in CAAX injected embryos. Either way, where is normal, endogenous *snai1b* expression observed? Please correct the methods to reflect that this was performed in CAAX injected embryos. If the GFP-CAAX fish are the guide for normal, endogenous expression, how do you define "ectopic", and why are there 20% of the GFP-CAAX fish with "ectopic" cells (Figure S4D)? This is still confusing and makes it very difficult to understand the results in the EGFP-KRASV12 fish.

We did compare ectopic expression in CAAX and KRas to uninjected embryos, as stated in the Methods. Endogenous expression of *snai1b* can be seen in the head and heart regions of 2 dpf zebrafish embryos in SFig. 4C, as well as by other groups (for example, see Fig. 6M of Giuliadori et al., Cardiovascular Research 2018). We originally submitted comparisons of total ectopic cells as lump sum numbers, as they made the statistics very difficult. However, we have now captured this in a way

that is more reflective of what we actually see in the revised graph in SFig. 4D: from 17 CAAX embryos, 2 presented < 5 'ectopic' cells each, and 0 with >5 'ectopic' cells, whereas from 22 KRas^{V12} embryos, 4 presented < 5 'ectopic' cells each, and 14 presented > 5 'ectopic' cells. This is a significant difference. We validated that these 'ectopic cells' were invaded cells, which is what we report in the confocal images.

5b. What has also not been answered from our initial questions, is whether all of the "ectopic" snai1b+ cells are double positive for snai1b and EGFP, and what is the percentage of double-positive cells compared to total invaded EGFP-positive cells? In Figure S5E, example images are shown of snai1b+/EGFP+ or EGFP+ only cells, but this is not quantified anywhere.

We could not quantify this as the DIG signal frequently (but not always, as in Fig. S5E) quenches GFP fluorescence.

In line 89 of the main text, it is stated "Additionally, ~15% of invaded cells express the mesenchymal marker, N-cadherin (Fig. 3H&I) and mesenchymal transcription factor, snai1b (Fig. S4C&D)." It is not clear if cells with ectopic snai1b expression belong to the same population as the N-cadherin positive cells from this sentence. Are 15% of invaded cells N-cadherin positive, another 15% snai1b positive? The quantification for snai1b is missing here, please amend the text.

The N-cadherin immunostaining and snai1b in situ hybridization data are from different experiments and we have amended the sentence, given that this may be misleading. Therefore, on line 89, we deleted the snai1b part of the sentence, as it is not important for the argument we make in the discussion.

5c. Minor comment: Anterior and posterior in Fig S4C might actually be dorsal and ventral. If so, this should be corrected.

Our apologies. The labels were moved inadvertently in the rearrangement of figure panels. They are now corrected in the resubmitted SFig. 4.

Reviewer #2 (Remarks to the Author):

The authors have addressed part of my concerns by providing new movies and further explanations. Most of the argument provided by authors in the text and the rebuttal to exclude alternative hypothesis

(leaky labelling, transport by macrophages) are overall very reasonable but I feel that most of them are still debatable (see below). I believe that the manuscript still lack some critical experiments that would really remove any ambiguity regarding the fate of the active Ras cells, including movies with nuclei labelling and long term fate tracking of cells. Because the fate of the Ras cells is the essential message of this article, I think it would really deserve proper fate mapping/tracking.

Since the latter is not possible through long term live imaging, and that the Cre system is not available now, I do believe there might be alternative based for instance on photoconversion of some epidermal Ras cells and localization/morphological description of the photoconverted cells one or two days later. This would solve the questions related to potential leakage of the construct and the origin of the internal cells. I do appreciate that this experiment will not be straightforward, but this would solve once for all the question of the fate of the Ras cells.

We recognize now that the previously requested data on misexpression versus invading cells only served to confound interpretation, making it seem that most invaded cells stem from misexpression. They do not. To clarify this, we have now quantified cells per fish into misexpression categories (muscle, notochord, or melanocyte,) **compared to** invaded cells in SFig. 1B. The new graph clearly shows that there are far more invaded cells than mis-expressing cells in UAS:EGFP-KRas^{V12}-injected into periderm-Gal4 but not in WT embryos. While there are more mis-expressing cells in the wild-type, we only found one potential invaded cell, indicating that invading cells **are not linked** to misexpression. To make it as clear as possible what we categorize as mis-expressing cells, we also provide examples of misexpression in the muscle, notochord, or melanocytes in SFig. 1C, as well as zoomed out examples of whole zebrafish tails from control wild-type or periderm-Gal4 fish (SFig. 1D). Invaded cells are those not fitting into these mis-expression categories or still remaining at the epidermis. We have updated the figure legend title for this in blue as well. While the Gal4 driver typically expresses fewer GFP-positive cells than does the KRT4 driver, used in the majority of the paper, it produces a similar high ratio of invaded to misexpressed cells. The similar number of mis-expressed cells stemming from KRT4:KRas^{V12} are far less than those expressed within the periderm (see Fig. 2C), further reducing the likelihood that any cells scored as 'invading' stem from mis-expression. We believe that this quite clearly shows that muscle or other cells that misexpress KRas^{V12} do not contribute to the invaded cells that we scored.

The suggestion of photoconverting or Cre-labelling cells would take well over a year and goes way beyond the findings we are presenting here: that cells invade and scrape off epithelial determinants by BCE. The long-term fate of invaded KRas^{V12} cells is indeed a very interesting question—and will be the subject of future papers.

Regarding the arguments provided by the authors regarding macrophage contribution in the rebutal, while they are all reasonable, I am not sure they totally exclude that Ras cell bodies movements is driven by them. For instance, in movie S9, while the GFP is clearly away from macrophage at the beginning, there is an overlap with purple signal (albeit fainter) when the GFP signal starts to move around.

We are quite confused by this query. The movie (S9) we show has a GFP cell that migrates down while a macrophage moves up in the opposite direction. Then after GFP-KRas cell moves away from the macrophage, the cell divides. Are you suggesting that the macrophage has engulfed the cell that migrated **away** from it, and then can divide after it is engulfed? This would be the first time we have ever heard of a dying cell dividing within a macrophage, especially one that it is distant from. Are you instead referring to a movie that doesn't have macrophages?

Similarly, the argument about the red labelling of macrophage (if they have already ingested before other labelled cells) is correct provided that the majority of macrophage have already engulfed labelled cells (which depends on the number of macrophage and the proportion of labelled cells). If there is a significant proportion of macrophage that have not yet engulfed labelled cells, the argument would not hold.

It is unclear what the question here is. If it is about immunostaining, it eliminates the issue of GFP loss by acidification. Thus, when we visualize ALL internalized cells and ALL macrophages using immunostaining, there is no significant overlap, which means that invaded cells **are not in** macrophages. Additionally, the fact that GFP would fade following ingestion (as seen with our example of a dying invaded cell that was engulfed) means that we would never have scored migrating GFP cells, if they were ingested in the first place.

Aside from these arguments, we have gone to considerable effort to address that invading cells that migrate, divide, and transdifferentiate cannot be a product of macrophage engulfment using five separate and rigorous methods:

- (1) They do not colocalize with mpeg:mCherry-labeled macrophages in live imaging.
- (2) They do not colocalize with macrophage markers in fixed samples stained with a chicken anti-GFP antibody, where GFP **does not** disappear from the acidification associated with phagocytosis, as they would in live movies. Importantly, if the GFP did disappear, we would never have scored them in the first place, so this is already a moot point.
- (3) For contrast, we show an example (SFig. 2a and SMovie 7) of an invaded cell that dies and IS ingested where GFP disappears, as expected. This dying cell's fragmentation, ingestion, and loss of GFP is **completely different** to what we see in all our invasion movies.

(4) We include several additional movies of green KRas^{V12} cells remaining for hours and migrating following extrusion—events that we could not have observed if macrophages engulf live invaded cells, like they do when cells die, since GFP disappears. You also need to consider: how might cells divide or differentiate, if dead (or even live) within a macrophage?

(5) We show that few of the immunostained invaded cells are caspase-3-positive, also ruling out death and engulfment as a mode for their later migration.

Regarding ectopic labelling, while the number of embryos with ectopic labelled cells is the same in the CAAX and the Ras embryo (Figure S1B), nothing exclude that the transfection of Ras + p53 deletion favor the survival and proliferation of the ectopic labelled cells (as pointed out by referee 1), which could explain the higher number of internal cells in this context. So while the interpretation proposed is plausible and reasonable, I don't think it is possible now to fully exclude the contribution of ectopic labelling unless there is a proper tracking and/or fate mapping.

This is not a “transfection/deletion” experiment. Expression plasmids containing EGFP-KRas^{V12} or CAAX were injected into one-cell embryos, and these cassettes integrated into the genome and were expressed from there, albeit not stably. Perturbation of p53 was done either with a loss-of-function point mutant or else knocked down with morpholino. As this point was already stated previously, we refer this reviewer to our previous response, above, and the revised graph in Fig. S1B. This very clearly shows, without having to resort to costly CRE-labelling that most KRas-transformed cells targeted to the periderm remain either within the periderm or invade, whereas those few misexpressed cells stay within the mis-expressed (usually muscle) cells.

Other minor points:

- Figure S1B: Could the authors provide details on how the transfected cells were tracked in absence of Gal4 ? I guess there was another fluorescent label transfected but this could help to specify in the legend. Also, could they provide more details on what they defined as misspecified (I suspect non-epidermal but this should be stated somewhere) ? Also I would have expected this number (proportion of embryo with ectopic cells) to be higher in RasV12 cells compared to CAAX cells (right part of the graph) specially if Ras promote cell internalisation. Actually this quantification could be compatible as well with Ras transfection promoting survival and proliferation of ectopic labelled cells (same initial number of ectopic labelled cells with the control, but higher cell number at the end because of higher proliferation and survival).

In the experiment where we injected UAS:EGFP-KRasV12 into WT (i.e. no Gal4 driver) embryos, we confirmed that injections were successful using a *cmlc2*:GFP reporter (GFP expression in heart cells), a component of our Tol2kit tools. As already mentioned above:

To make it as clear as possible what we categorize as mis-expressing cells, we also provide examples of misexpression in the muscle, notochord, or melanocytes in SFig. 1C, as well as zoomed out examples of whole zebrafish tails from control wild-type or periderm-Gal4 fish (SFig. 1D). Invaded cells are those not fitting into these mis-expression categories or still remaining at the epidermis. While the Gal4 driver typically expresses fewer GFP-positive cells than does the KRT4 driver, used in the majority of the paper, it produces a similar high ratio of invaded to misexpressed cells. The similar number of mis-expressed cells stemming from KRT4:KRas^{V12} are far less than those expressed within the periderm (see Fig. 2C), further reducing the likelihood that any cells scored as 'invading' stem from mis-expression.

- It was not clear to me where were the invading cells in movie S4 among all the GFP basal round bodies. Could the authors point to the relevant structure in the movie?

We now point to the one of the invading cells in Movie S4.

- Figure S2C right: could the authors indicate on the panel where are the extruded Ras cells ?

In SFig. 2C, we now mark the cells that remain peridermal; all other GFP+ cells have invaded.

- When the authors refer to the neuron like morphology, they do observe rare case of neuron like morphology in the CAAX control. Would that be a typical example of ectopic labelling ?

No. Most misexpression occurs within muscle cells, with some melanocytes and notochord, as described above in our response to your first question. As indicated in the graph on Fig. 3G, in the 57 embryos scored, we only found one neuronal-like cell in embryos injected with the CAAX control.

- Figure 3H: what are these other labeled N-cad structures which are not GFP positive ?

Embryos contain cells that express N-cadherin during development and we presume that these are such cells.

- Figure 4F: Since there is no clear evidence that the purple label structure at the bottom comes from the extruding cell (purple arrow) I would not indicate it on the figure and the legend unless the authors can unambiguously track it (which I guess they cannot if it is out of the frame for several time frame).

This is a reasonable request, as we cannot unambiguously tell that this came from the basally extruding cell, as we can in many other movies, and so have removed it.

Reviewer #3 (Remarks to the Author):

This study is novel and interesting to a broad readership. It is of a sufficient standard to warrant publication at Nature Communications.

We thank the reviewer for their comments which helped improve the manuscript.

Thus, together, we have addressed all of the comments raised by the reviewers in ways that we think are far more rigorous than what they have suggested. We have fixed the confusing graphs and added examples to SFig. 1 that previously raised concerns about the origin of the invaded cells. These findings indicate that the few cells misexpressing KRas cells do not contribute to internalized cell populations. We also provide a large array of strong evidence that the live, invading cells are not phagocytosed by macrophages, in contrast to the nominal dying ones that are, and that E-cadherin is apical, near the apical cortical actin ring, where it can get scraped off with BCE.

Sincerely,

Prof. Jody Rosenblatt

REVIEWERS' COMMENTS

Reviewer #2 (Remarks to the Author):

The authors have clarified their quantifications regarding misexpressing cells. It seems as such clear that there are more invading cells in Ras context and that it is unlikely explained by leaky expression. As for the long term fate of basally extruding cells, I understand that other methods for long term tracking cannot be applied without much more work and this may be the focus of other studies.

To clarify, my doubts were coming from the combination of these two points : potential leakiness and movement driven by macrophage of cell debris. While the first point would have explained the inner cell mass and the ectopic expression, the later could explain the movement of GFP positive structure shown in the movie. Since the leakiness problem is solved, it is clear that the engulfment of dying Ras cells by macrophage cannot explain these inner cell mass and the occurrence of division and this point becomes less critical.

I still believe that movies with a nuclear marker will help to sort out apoptotic cells from surviving ones and indeed the movie provided in their bioarchive manuscript (S9) is rather convincing. If possible, it may help to add this data in the manuscript.

To conclude, while further experiment (long term tracking and/or fate mapping) would help to unambiguously document the fate of basally extruding cells, I believe that the novelty of these observations still justify publication.

Reviewer #4 (Remarks to the Author):

In the immunostaining data, the authors showed that E-cadherin only localized in the apical domain of the periderm cells. However, as stated by the reviewer 1, the distribution of E-cadherin proteins in periderm cells has been studied quite extensively, and it was shown (using the same antibody with this manuscript) clearly that E-cadherin are present in the basolateral domain (Arora et al. *eLife* 2020;9:e49064; RübSam et al., *Nat Commun.* 2017; 8: 1250). These data look clear and convincing.

In the rebuttal letter, the authors cited several other references to support their claim that apical-localization of E-cadherin is more widely accepted in the field. However, these studies did not claim that E-cadherin is absent in the basolateral domain. Finally, the author also argued that basolateral localization of E-cadherin cannot be true because it would prevent the formation of intimate contact between epidermal cells. However, I disagree with the authors. Unlike tight junction proteins, which

are mainly expressed in the outmost epidermal cells and distributed in the apical-lateral domain, E-cadherin is expressed by the epidermal cells in various layers and is important for their interactions with both lateral and vertical neighbor cells.

In my opinion, the authors should carefully repeat or reanalyze their data, perhaps using a better apical marker. Even if their results are repeatable, the authors should modify their model to include the integration of additional mechanisms to explain the absence of basolateral E-cadherin in their model.

Responses to the Reviewers:

Reviewer #2 (Remarks to the Author):

The authors have clarified their quantifications regarding misexpressing cells. It seems as such clear that there are more invading cells in Ras context and that it is unlikely explained by leaky expression. As for the long term fate of basally extruding cells, I understand that other methods for long term tracking cannot be applied without much more work and this may be the focus of other studies.

To clarify, my doubts were coming from the combination of these two points : potential leakiness and movement driven by macrophage of cell debris. While the first point would have explained the inner cell mass and the ectopic expression, the later could explain the movement of GFP positive structure shown in the movie. Since the leakiness problem is solved, it is clear that the engulfment of dying Ras cells by macrophage cannot explain these inner cell mass and the occurrence of division and this point becomes less critical.

I still believe that movies with a nuclear marker will help to sort out apoptotic cells from surviving onr and indeed the movie provided in their bioarchive manuscript (S9) is rather convincing. If possible, it may help to add this data in the manuscript.

To conclude, while further experiment (long term tracking and/or fate mapping) would help to unambiguously document the fate of basally extruding cells, I believe that the novelty of these observations still justify publication.

Thanks for the reviews. We are happy to provide the supplementary movie 6, which we think you are referring to (not 9, which is already present) from the bioarchive to provide the nuclear marker. We have added this and refer to it as Supplementary Movie 9.

Reviewer #4 (Remarks to the Author):

In the immunostaining data, the authors showed that E-cadherin only localized in the apical domain of the periderm cells. However, as stated by the reviewer 1, the distribution of E-cadherin proteins in periderm cells has been studied quite extensively, and it was shown (using the same antibody with this manuscript) clearly that E-cadherin are present in the basolateral domain (Arora et al. eLife 2020;9:e49064; Rübsam et al., Nat Commun. 2017; 8: 1250). These data look clear and convincing.

In the rebuttal letter, the authors cited several other references to support their

claim that apical-localization of E-cadherin is more widely accepted in the field. However, these studies did not claim that E-cadherin is absent in the basolateral domain. Finally, the author also argued that basolateral localization of E-cadherin cannot be true because it would prevent the formation of intimate contact between epidermal cells. However, I disagree with the authors. Unlike tight junction proteins, which are mainly expressed in the outmost epidermal cells and distributed in the apical-lateral domain, E-cadherin is expressed by the epidermal cells in various layers and is important for their interactions with both lateral and vertical neighbor cells.

In my opinion, the authors should carefully repeat or reanalyze their data, perhaps using a better apical marker. Even if their results are repeatable, the authors should modify their model to include the integration of additional mechanisms to explain the absence of basolateral E-cadherin in their model.

You have pointed out some interesting papers that have made us revise our interpretations. We believe that the point you raise about E-cadherin being basolaterally localized in stratified epithelial layers make sense, as shown in RübSam et al., Nat Commun. 2017; 8: 1250, as it would help adhere different epithelial layers. This basolateral distribution reported in the Arora et al paper may occur at earlier stages than we investigated or may be basolateral compared to the other markers they used, rather than actin, which is our focus here. In re-analyzing our zebrafish images, we confirm that all the E-cadherin in the periderm co-localizes with apical cortical actin, pertinent to our findings here. We have added a movie (Supplementary Movie 17) that slowly focuses through slices at high resolution in red and green, where the overlap is easier to visualize, clearly demonstrating its colocalization with cortical, apical actin.

Because E-cadherin colocalizes with cortical apical actin throughout the periderm (yet not the underlying layer), we have kept this localization in the schematic but smeared it out a bit with the actin, as seen in our micrographs. Additionally, we now discuss how tissues that have basolaterally-localized E-cadherin may not scrape off epithelial determinants during basal extrusion. We also raise the point that if cell escape by apical extrusion, they may also retain E-cadherin. While it is not clear if tumours can spread by apical extrusion, it could account for how pancreatic cancer spreads via the duct it shares with the liver.

We have added this sentence to our results/discussion on lines 181-88:

While in our studies, E-cadherin and ZO-1 predominantly localize with cortical, apical actin and are stripped off by BCE, it is important to note that cells invading from epithelia where E-cadherin also localizes basally, such as stratified epithelia^{38,39}, may not lose their epithelial determinants. Additionally, cells could escape by extruding apically into the duct, in which case, we would expect them not to lose apically localized epithelial determinants as they invade. Future work will need to determine if

cells invading with E-cadherin lose expression over time through different mechanisms or if the mechanism of invasion impacts later fate and tumour aggressiveness.

We hope that this addresses this point well, as we think it is an important point that may vary depending on how and where cells extrude from.